# Human cortical encoding of pitch in tonal and non-tonal languages

Yuanning Li [1,2,7], Claire Tang[1,2,7], Junfeng Lu [3,4,7], Jinsong Wu[3,4,5,6 ✉] & Edward F. Chang [1,2 ✉]

Languages can use a common repertoire of vocal sounds to signify distinct meanings. In tonal languages, such as Mandarin Chinese, pitch contours of syllables distinguish one word from another, whereas in non-tonal languages, such as English, pitch is used to convey intonation. The neural computations underlying language specialization in speech perception are unknown. Here, we use a cross-linguistic approach to address this. Native Mandarin- and English- speaking participants each listened to both Mandarin and English speech, while neural activity was directly recorded from the non-primary auditory cortex. Both groups show language-general coding of speaker-invariant pitch at the single electrode level. At the electrode population level, we find language-specific distribution of cortical tuning parameters in Mandarin speakers only, with enhanced sensitivity to Mandarin tone categories. Our results show that speech perception relies upon a shared cortical auditory feature processing mechanism, which may be tuned to the statistics of a given language.

[1] Department of Neurological Surgery, University of California, San Francisco, CA, USA. [2] Center for Integrative Neuroscience, University of California, San Francisco, CA, USA. [3] Brain Function Laboratory, Neurosurgical Institute of Fudan University, Shanghai, China. [4] Shanghai Key laboratory of Brain Function Restoration and Neural Regeneration, Shanghai, China. [5] Neurologic Surgery Department, Huashan Hospital, Shanghai Medical College, Fudan University, Shanghai, China. [6] Institute of Brain-Intelligence Technology, Zhangjiang Lab, Shanghai, China. [7] These authors contributed equally: Yuanning Li, Claire Tang, Junfeng Lu. ✉email: wujinsong@huashan.org.cn; edward.chang@ucsf.edu

Pitch is used in all languages for conveying prosodic information. In tonal languages such as Mandarin Chinese, the modulation of pitch called lexical tones also conveys differences in word meanings. Mandarin has four distinct tones: high-level (T1), mid-rising (T2), low-dipping (T3), and high-falling (T4)[1]. A monosyllable /ma/ can mean /ma1/ (mother 妈), /ma2/ (hemp 麻), /ma3/(horse 马), and /ma4/ (to scold 骂) depending upon which tone is used. The primary acoustic cue for lexical tone is fundamental frequency (F0), perceived as pitch[2].

Listeners extract lexical tone information despite significant sources of pitch variability. In natural speech, the actual pitch characteristics of lexical tones for Mandarin words are highly variable both across speakers, determined by the anatomical properties of the speaker's vocal folds and larynx, and across utterances, affected by the discourse prominence of words, sentence-level phrasing, and the downdrift of fundamental frequency across an utterance[3,4]. The existence of cross-speaker and cross-utterance variation necessitates the normalization of pitch for lexical tone perception[4–6].

Whole-brain neuroimaging studies[7–13] have consistently localized lexical tone processing to the human non-primary auditory cortex in the bilateral superior temporal gyri (STG)[14]. The STG is a critical locus for phonological processing and has been shown to contain selective encoding for a large diversity of phonetic features[15]. However, a more fundamental question than anatomical localization is how tone information is encoded by cortical activity–that is, the precise mapping between stimulus features and neural responses. What kind of information is represented in the STG is unknown: the acoustic absolute pitch frequency (F0)[16,17], a high-level auditory representation such as speaker-normalized pitch[18], or the abstract linguistic lexical tone identity[10]? Furthermore, is the encoding of tone-related pitch specialized in Mandarin speakers, compared with those who do not speak a tonal language? Understanding the neurological basis of tone processing can address fundamental questions about how specializations in the human auditory system transform acoustic cues into meaningful linguistic percepts.

In this work, to address these questions, we record neural activity from 15 participants (11 native Mandarin speakers, and 4 native English speakers; all are monolingual speakers) undergoing neurosurgical brain mapping procedures[19]. Cortical neural activity is recorded using high-density electrodes arrays (electrocorticography, ECoG), which are located on the lateral surface of the exposed temporal lobe, whereas participants passively listen to natural, continuous, Mandarin[20], and English speech[21,22]. Direct high-density cortical recordings are necessary to resolve detailed neural activity, spatially for selective tuning to acoustic features on the scale of millimeters, and temporally for tracking responses to pitch dynamics at tens of milliseconds. By using the same stimuli for both Mandarin-speaking and English-speaking participants in a cross-linguistic paradigm, our goal is to address how the human brain processes sound pattern variability within and across languages. This provides new fundamental insights into the shared potential mechanisms of auditory processing in tonal and non-tonal languages in humans.

## Results

### Speaker-normalized pitch defines lexical tones in Mandarin.
The acoustic waveforms and spectrograms of an example Mandarin sentence spoken by a female and a male native Mandarin speaker are shown in Figs. 1a and 1b, respectively. Each syllable in Mandarin has a lexical tone that is primarily cued by the auditory feature of pitch, but the absolute pitch values of a Mandarin tone vary from speaker to speaker owing to the speaker's baseline pitch (Figs. 1c, 1d). Figure 1e shows the average absolute pitch contour

for each lexical tone for each of the ten speakers in the stimulus corpus. The contours occupied different regions of the absolute pitch space, such that no absolute pitch contour determined tone identity (Fig. 1e). In order for pitch information to be linguistically useful, it must be invariant to speaker identity. Two pitch features that have been shown to be important in tone perception are speaker-normalized relative pitch height and pitch change[23–26]. We examined these two features in our stimulus corpus. The contours for both relative pitch and pitch change for every tone were consistent between speakers and distinct from each other (Fig. 1f–k). Moreover, a principal component analysis (PCA) of the pitch contours of tone tokens in the corpus also revealed similar key features (Supplementary Fig. 1). Furthermore, substantial within-tone variance was still present even after speaker-normalization (Figs. 1f, 1g and Supplementary Fig. 1), presenting a classification problem to listeners who need to reliably extract tone identity.

### Relative pitch encoding underlies tone discrimination in STG.
To understand how lexical tones are represented in the human non-primary auditory cortex, we first evaluated the neural activity from the 11 native Mandarin-speaking participants while they passively listened to Mandarin speech (see Supplementary Fig. 3 for grid placement in each participant). We computed the amplitude of signals in the high-gamma band (70–150 Hz), a measure correlated with local neuronal activity[27]. We found widespread cortical activation in STG in response to Mandarin speech, with 49 (±19 s.d.) speech-responsive electrodes per subject on average (Fig. 2a and Supplementary Figs. 2–3), similar to previous observations in English-speaking participants[15].

Overall, we observed clearly discriminable response patterns to different tones. However, we did not see evidence for electrodes that responded only to one particular tone category. Figure 2a shows electrodes that have significantly different responses to lexical tones (F test, $p < 0.05$, two-sided, Bonferroni corrected) for one participant, where high-gamma responses were aligned to vowel onsets in each syllable. Tone-discriminating electrodes made up an average of 16% (±7% s.d.) of speech-responsive electrodes (Supplementary Figs. 2–3). Figure 2b, g shows the average neural response to each tone for two example STG electrodes. The first electrode showed higher responses to tone 1, 2 and 3. The second electrode responded most to tones 1 and 4. This shows that adjacent STG electrodes may be differentially tuned to lexical tones.

Instead of discrete tone identity encoding, we found that single-electrode neural responses differentiating lexical tones were best explained by speaker-normalized pitch features of relative pitch height and pitch change. We fitted encoding models (Supplementary Fig. 4) to test whether these complex pitch features could predict neural activity beyond absolute pitch height and discrete tone identity. We first looked at model predictions for individual electrodes to understand how encoding of speaker-normalized pitch features leads to tone discriminability. The actual responses and model predictions of the two example STG electrodes are shown in Fig. 2b–e and Fig. 2g–j. Spectrum, intensity, and absolute pitch captured the average dynamics of neural responses but did not predict differences between lexical tones (Electrode 1: unique $R^2 = 0.072$, 5e-4, -3e-5; Electrode 2: unique $R^2 = 0.042$, −0.002, −0.002, for spectrum, intensity and absolute pitch respectively; Figs. 2c, h). Furthermore, a larger encoding model that included predictors for tone category did not fully recover the actual differential patterns in the neural activity, either (Electrode 1: tone category unique $R^2 = 0.007$, $p < 0.05$, permutation test; Electrode 2: unique $R^2 = −0.002$, $p > 0.5$, permutation test; Figs. 2d, 2i).

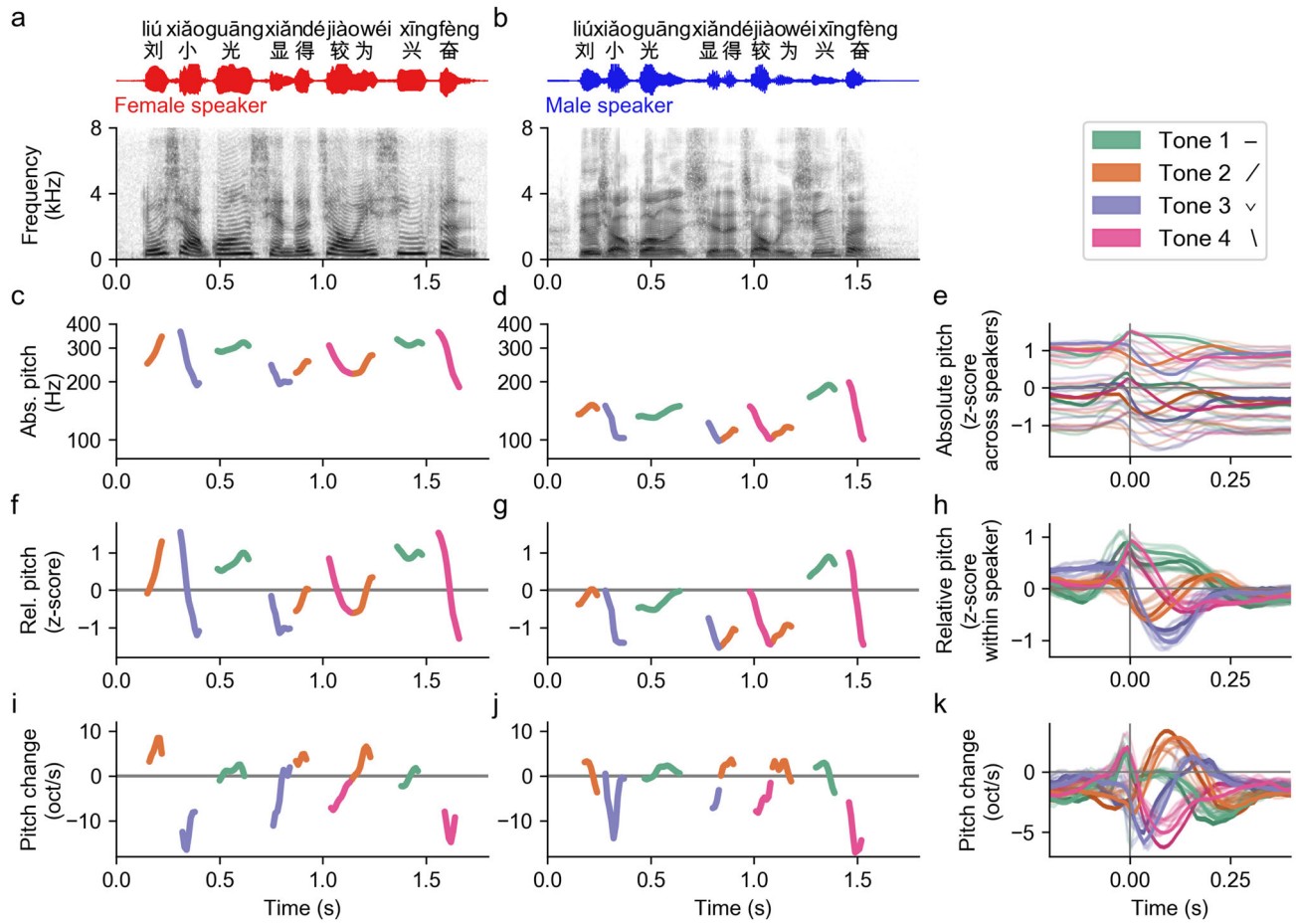

**Fig. 1 Speaker-normalized relative pitch and pitch change are key acoustic features for lexical tones in continuous Mandarin speech. a**, **b** Acoustic waveform (top) and spectrogram (bottom) for two example sentences (transcript in Chinese and Pinyin) spoken by a female and a male speaker, respectively. **c**, **d** The absolute pitch of the two example sentences. Each syllable is colored according to its lexical tone. **e** Average absolute pitch contours for each lexical tone aligned to vowel onset for the ten speakers in the speech corpus (five females, five males). The two speakers of the example sentences are shown in thicker lines. The lighter colored traces are from the female speaker and the darker colored traces are from the male speaker. The absolute pitch contours from the other speakers are shown more transparently. **f**, **g** The relative pitch contours of the example sentences, colored by the lexical tone of each syllable. Relative pitch is calculated by normalizing pitch values by speaker (z score of log Hz). **h** Average relative pitch for each tone aligned to vowel onset for all ten speakers. **i**, **j** Pitch change of the example sentences, colored by the lexical tone of each syllable. **k** Average pitch change for each tone aligned to vowel onset for all ten speakers.

However, with the addition of speaker-normalized features to this model, predicted neural responses matched actual responses to the four lexical tones (Electrode 1: speaker-normalized pitch unique $R^2 = 0.037$, $p < 0.05$, permutation test; Electrode 2: unique $R^2 = 0.016$, $p < 0.05$, permutation test; Figs. 2e, j). The regression weights for the two example electrodes indicate selectivity for positive pitch change and negative pitch change, respectively (Figs. 2f, k). Across all tone-discriminating electrodes, we saw a variety of other tuning patterns, including selectivity for low relative pitch height (Supplementary Fig. 5a), and high relative pitch height (Supplementary Fig. 5b). The defining similarity was the encoding of speaker-normalized pitch features, rather than absolute pitch or lexical tone categories.

We then quantified encoding across all STG electrodes to test which acoustic features were encoded by electrodes that discriminated between tones. We were particularly interested in comparing the encoding of speaker-normalized pitch features, absolute pitch and lexical tone categories on these electrodes. Out of 541 speech-responsive electrodes across all 11 native Mandarin-speaking participants, speaker-normalized pitch features uniquely explained a significant amount of variance (unique $R^2$ up to 6%) on 112 (20.7%) electrodes (Fig. 3a). There were separate, fewer

sites in STG that encoded absolute pitch (42, 7.8%), and absolute pitch encoding explained significantly less variance in neural activity (up to 2%, with unique $R^2$ above 1% in only one electrode; $t(129) = -5.84$, $p = 4 \times 10^{-8}$, paired two-sided $t$ test, Fig. 3a). Moreover, including lexical tone category as discrete predictors in the encoding model also explained little unique variance (only up to 1.5%), significantly less compared with speaker-normalized pitch features ($t(151) = -5.46$, $p = 2 \times 10^{-7}$, paired two-sided $t$ test, Fig. 3b).

Among electrodes tuned to speaker-normalized pitch features, individual electrodes were tuned to either relative pitch height or pitch change, with few electrodes showing strong tuning to both features (Fig. 3c). Furthermore, the variance explained by speaker-normalized pitch features was significantly correlated with single-electrode between-tone discriminability (quantified as the maximum $F$-statistics across the 1 s time window after tone onsets; Pearson's $r = 0.85$, $n = 112$ electrodes, $p = 9 \times 10^{-32}$; Fig. 3d) whereas the variance explained by absolute pitch was not (Pearson's $r = 0.17$, $n = 42$ electrodes, $p = 0.3$; Fig. 3e). Therefore, at the level of individual electrodes, the differential neural responses with regard to lexical tones are mainly driven by the encoding of speaker-normalized pitch features rather than absolute pitch or

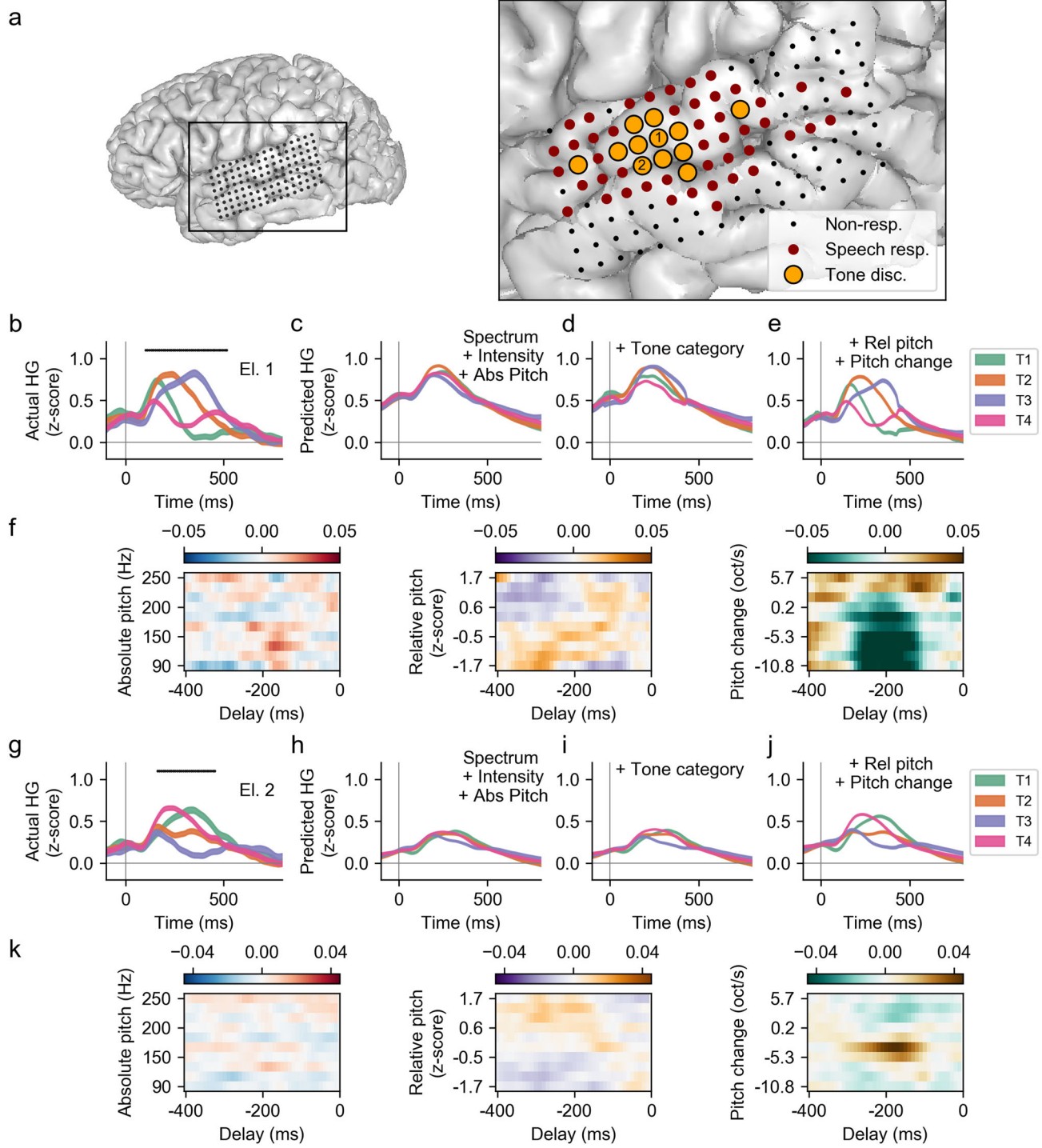

**Fig. 2 Mandarin lexical tones evoke differential neural responses at single STG electrodes, explained by tuning to speaker-normalized pitch features.**
**a** Electrode locations for one participant. Dark red indicates electrodes that were responsive to speech. Circled orange electrodes had significantly different response patterns to different tones. **b** The actual high-gamma (HG) responses from a single example electrode (Electrode 1 in **a**) that differentiates lexical tones. Average neural activity (shaded area represents mean ± s.e.m.) in response to each tone is plotted, aligned to vowel onset. Black lines indicate time points where means are significantly different between tones (P < 0.05, F test, two-tailed, Bonferroni corrected). **c** The predicted responses from an encoding model using spectrum, intensity and absolute pitch features. **d** The predicted responses by adding tone category features to the model in **c**. **e** The predicted responses from the full encoding model where speaker-normalized pitch features are included. **f** The temporal receptive field (regression weights) from the model with regard to absolute pitch, relative pitch height and pitch change, indicating selectivity to positive pitch change. **g**–**k** Same as **b**–**e**, but for Electrode 2, which is tuned to negative pitch change.

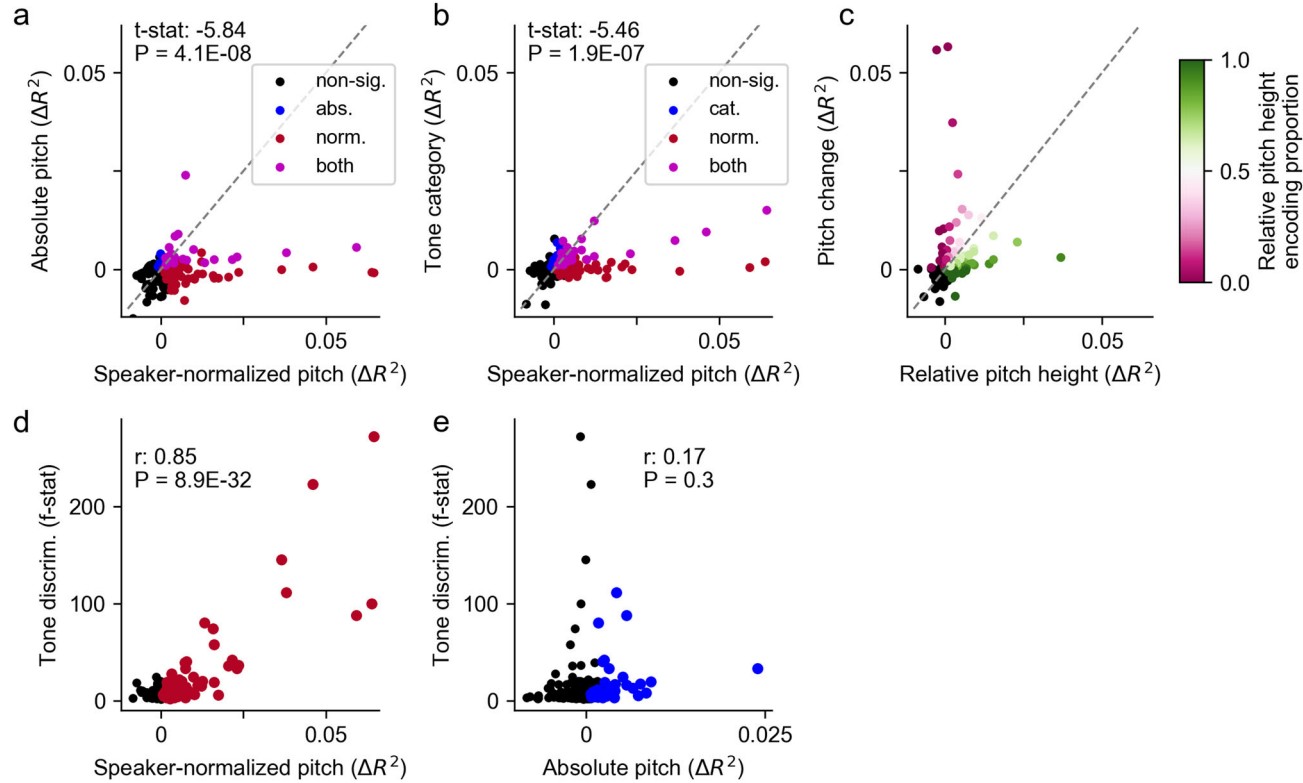

**Fig. 3 Encoding of speaker-normalized relative pitch height and pitch change, but not tone category or absolute pitch, underlies lexical tone discriminability at individual electrodes. a** Scatterplot of the unique variance explained by absolute pitch and by speaker-normalized pitch features, across speech-responsive electrodes from all Mandarin-speaking participants. Each dot represents a single-electrode. Colored dots indicate significant encoding of absolute pitch only (blue), speaker-normalized pitch only (red) or both (magenta). The $t$ statistic is computed using paired $t$ test between absolute pitch ($R^2$) and speaker-normalized pitch ($R^2$). **b** Scatterplot comparing the relationship between the unique variance explained by speaker-normalized pitch features and discrete tone category features in the full encoding model in single electrodes. Colored dots indicate significant encoding of tone category only (blue), speaker-normalized pitch only (red) or both (magenta). The $t$ statistic is computed using paired $t$ test between-tone category ($R^2$) and speaker-normalized pitch ($R^2$). **c** Scatterplot of the unique variance explained by relative pitch and by pitch change in single electrodes. Colored dots indicate significant encoding of either set of features, with the color indicating the proportion of the variance explained by the corresponding feature set. **d** Unique variance explained by speaker-normalized pitch features and tone discriminability. Colored dots indicate electrodes that had significant encoding of speaker-normalized pitch features ($p < 0.01$, permutation test). The $r$ value is computed using Pearson's correlation between-tone discriminability ($F$ value) and speaker-normalized pitch ($R^2$). **e** Unique variance explained by absolute pitch and tone discriminability. Colored dots indicate electrodes that had significant encoding of absolute pitch ($p < 0.01$, permutation test). The $r$ value is computed using Pearson's correlation between-tone discriminability ($F$ value) and absolute pitch ($R^2$). $P$ values in (**a**, **b**, **d**, **e**) are computed using two-sided $t$ test.

other lower-level acoustic features (Supplementary Fig. 6). Hence, our results address outstanding questions from previous behavioral and neurophysiological studies, which have shown that speaker-normalized pitch features, both relative pitch height and pitch change, are important for tone perception[4,23–26,28,29]. Moreover, when similar analysis was applied to 183 speech-responsive STG electrodes in native English speakers when they listened to Mandarin speech (Supplementary Fig. 7), we also found significant tuning to speaker-normalized pitch height and pitch change, other than absolute pitch, suggesting a general auditory mechanism for pitch across speakers with different language experience.

**Speaker-normalized pitch encoding is shared across languages.** We next wanted to determine, for the same listener, whether the processing of speaker-normalized pitch is language-specific, or whether it can be explained by general, language-invariant auditory mechanisms for pitch processing. To do this, the same Mandarin-speaking participants also listened to English speech (Fig. 4a), which has pitch variation for stress and intonation but not for tone. Thus, we tested whether speaker-normalized pitch was also extracted when pitch variations did not signify lexical

content or perceived as tones. We fitted encoding models to predict neural responses from acoustic features based solely on the responses to English speech (English data). We then tested the ability of these models trained on English data to predict neural responses to Mandarin speech. Figure 4b shows the actual high-gamma responses to different tones for one example STG electrode. Figure 4(c, d) shows model-predicted neural responses from models fitted on Mandarin data and English data, respectively. Both models predicted qualitatively similar responses (Pearson's $r = 0.90$, between the two model predictions, total number of time points $n = 159,000$, $p < 1 \times 10^{-10}$), which captured the actual response pattern well (Mandarin model Pearson's $r = 0.58$, English model Pearson's $r = 0.54$, both have total number of time points $n = 159000$, $p < 1 \times 10^{-10}$; Fig. 4b–d). Although there is a performance gap between models (Supplementary Fig. 10g–i), the English data-trained model is largely able to capture the response patterns to tones in Mandarin.

To determine whether speaker-normalized pitch tuning underlies the ability of the English model to predict responses to tones, we compared the regression weights from the Mandarin and English models. Figure 4e, f shows the temporal receptive field (TRF) for speaker-normalized pitch features for the Mandarin

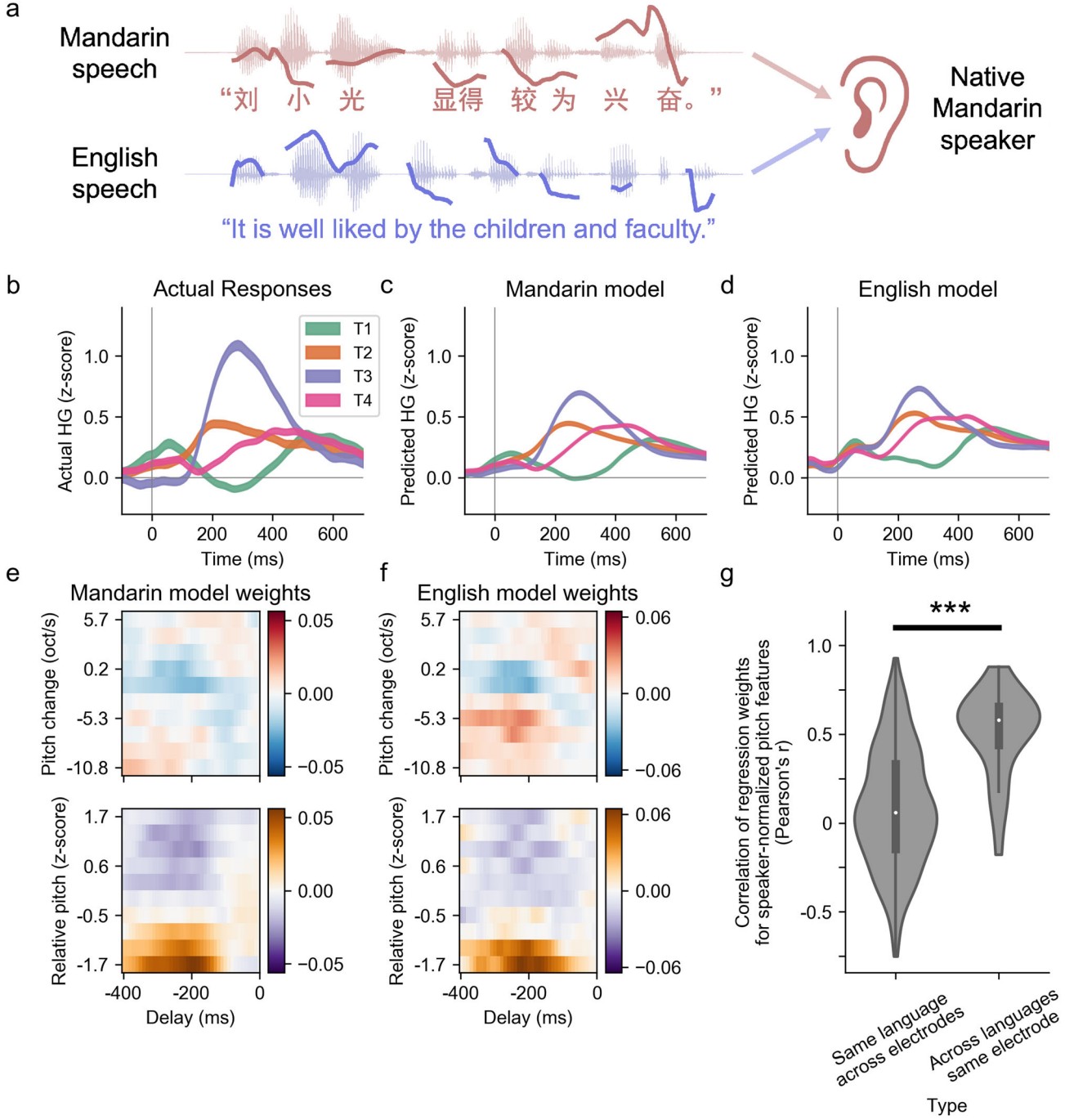

**Fig. 4 Single-electrode encoding of speaker-normalized pitch is language-independent. a** Participants listen to both Mandarin and English speech. Two encoding models are each trained using the neural responses to one language. **b–f** Representative example from one electrode. **b** Actual average evoked neural response to each lexical tone. **c** Predicted neural responses to each tone using the Mandarin encoding model, which is fitted on neural responses to Mandarin speech. **d** Predicted neural responses using the English encoding model, which is fitted on neural responses to English speech, showing a similar response pattern to **b** and **c**. **e** Regression weights for speaker-normalized pitch features over time from the Mandarin model whose predictions are shown in **c**. **f** Regression weights for speaker-normalized pitch features over time from the English model whose predictions are shown in **d**. The tuning to pitch is similar regardless of the stimulus language. **g** Comparing the empirical distributions of the correlation values between the regression weights for speaker-normalized pitch features. Central white dots represent the median values, and the thick vertical gray bars indicate interquartile range (first and third quantiles). The "across languages same electrode" comparison (right) shows significant positive correlations between the weights from the Mandarin model and English model within the same tone-discriminating electrode. The baseline distribution of the "same language across electrodes" comparison (left) shows no significant correlation between the regression weights from the Mandarin model of different tone-discriminating electrodes (***$p < 0.001$, two-sided Mann–Whitney $U$ test).

model and English model, respectively, corresponding to the models whose predictions are shown in Fig. 4c, d. They both showed consistent tuning for low relative pitch height (Pearson's $r = 0.88$, $n = 800$, $p < 1 \times 10^{-10}$). To quantify this similarity across all tone-discriminating neural populations, we calculated the correlation between the regression weights for the Mandarin and English models within the same tone-responsive electrode (Fig. 4g). For nearly all electrodes, the correlation was positive, indicating that the tuning for speaker-normalized pitch features was similar for both Mandarin and English in the same neural population. Furthermore, these correlations were significantly higher than the baseline statistical distribution drawn from correlations between different electrodes within the same participants (Fig. 4g; $U = 6.37$, $p = 1.9 \times 10^{-10}$, $n = 465$ total correlations, Mann–Whitney $U$ test, two-sided). Behavior evidence suggests that pitch normalization for tone perception may be based on general auditory mechanisms[6]. These results suggest that, for a given listener, the encoding of pitch at single electrodes is largely language-independent, and as such, provide further evidence for the local neural encoding of auditory pitch cues, and not the processing of lexical tone category.

**Neural tuning properties are shaped by language experience**. We next wanted to understand whether encoding is affected by language experience. While both languages use pitch, the statistics are different. Mandarin tone contours have a larger dynamic range of pitch[30,31] and are temporally organized by syllable-sized segments compared with English speech[32]. To address this, we had a control group of native English-speaking participants listen to the same Mandarin speech corpus. We compared tuning properties in STG between English and Mandarin speakers (Fig. 5a) to determine the differences in local encoding of speaker-normalized pitch, and to assess the implications of those local differences for population-level coding of tone category.

The tuning properties at each individual electrode with respect to speaker-normalized pitch features (relative pitch height and pitch change) were quantified by the modulation depth of the tuning curve (difference between the maximum and minimum high-gamma response) and the linearity of the tuning curve (Fisher transformed correlation coefficient between the pitch feature and high-gamma activity). Mandarin speakers showed a wider dynamic range that covered a balanced distribution of tuning for relative pitch height, whereas English speakers showed an asymmetric distribution biased towards tuning for high relative pitch (Fig. 5b, $p < 0.05$, permutation test). Neural responses in both native Mandarin speakers and native English speakers showed similar tuning for pitch change (Fig. 5d, $p > 0.5$, permutation test). This may reflect the relative importance of low pitch height for the contrastive perception of the low-dipping Tone 3 in Mandarin. Whereas in English, prosodic intonation primarily relies on high relative pitch, for example, when a given word is stressed in a sentence.

Furthermore, considering the temporal dynamics of lexical tones, different integration time may be necessary for processing the dynamics of the pitch contour in Mandarin speakers. To evaluate the integration time, we compared the average TRF of speaker-normalized pitch features of individual STG electrodes. Specifically, we computed the average absolute regression weight across relative pitch features for each time point, which represents the dynamic of the encoding properties in time. Neural responses from English speakers showed a more transient TRF for pitch height, peaking at ~100 ms, whereas Mandarin speakers had a far longer TRF that extended to ~300 ms (significantly higher than English speakers between 180 and 270 ms, paired $t$ test, two-sided, $\alpha < 0.05$, false-discovery rate corrected; Fig. 5c). This time scale of Mandarin TRF

is consistent with the average duration of tone contours in Mandarin speech[32]. No significant difference was found in the TRF for pitch change (Fig. 5e, $p > 0.5$, paired $t$ test). Therefore, although a general auditory coding for pitch is found at single electrodes, the relative distribution of encoding parameters can be influenced by language experience. Mandarin speakers showed a balanced distribution of positive and negative relative pitch tuning and longer integration time for speaker-normalized pitch contour, suggesting tuning to the specific statistics of Mandarin.

These results may have implications for interpreting results from speech perception studies. Namely, they predict that all listeners process speaker-normalized pitch, and therefore even non-tonal language speakers can perceive the psychophysical boundaries between lexical tones[33,34]. However, studies also suggest that native tonal language speakers may have enhanced sensitivity to linguistic boundaries beyond acoustic cues[33,35].

**Neural population sensitivity to tone categories**. Mandarin speakers are more sensitive to linguistically relevant tone contours while English speakers mainly perceive psychophysical pitch levels[33–35]. This suggests perceptual categorization in Mandarin speakers. To address this, we then examined whether there is enhanced sensitivity to tone categories that is reflected in the collective population of STG responses in Mandarin speakers.

Specifically, we pooled speech-responsive electrodes in the non-primary auditory cortex across subjects in this analysis (316 in Mandarin-speaking participants and 171 in English-speaking participants). We computed the peak overall pair-wise classification accuracy between the concatenated population neural responses to the four lexical tones using a multivariate pattern classifier[36], and compared with the tone classification accuracy applied to the acoustics itself. Compared with acoustic classification accuracy, the neural classification accuracy in Mandarin speakers was 28% higher over chance level ($t(198) = 7.88$, $p = 2.2 \times 10^{-13}$, bootstrapped two-sample $t$ test, $n = 200$ repetitions, two-sided; Fig. 5f), whereas no difference was observed in English speakers (4.2% change from acoustic space, $t(198) = 1.09$, $p = 0.28$, bootstrapped two-sample $t$ test, $n = 200$ repetitions, two-sided; Fig. 5f). Pitch-encoding electrodes contributed to the between-tone classification in both native Mandarin speakers and English speakers (Supplementary Fig. 13). Moreover, removing the strong negative pitch height encoding electrodes in Mandarin speakers (gray rectangles in Fig. 5b) from the analysis resulted in a significant reduction in tone decoding accuracy ($t(198) = 8.67$, $p = 1.6 \times 10^{-15}$, bootstrapped two-sample $t$ test, two-sided; Fig. 5f) and resulted in comparable accuracy to acoustic space ($-2.8\%$ change from acoustic space, $t(198) = -0.77$, $p = 0.44$, bootstrapped two-sample $t$ test, two-sided; Fig. 5f), which indicates that a balanced distribution with strong tuning in both positive and negative directions is critical for tone category representation.

We further wish to investigate the fundamental question about whether population-level neural response encodes this pitch variance along with the continuous parameters of acoustics (veridical representation of speaker-normalized pitch) or is biased towards the perceptual categories of tones. Particularly, we want to test if the neural response in STG is sensitive to tone categorical boundaries along acoustic feature dimensions, such as pitch height and pitch change. To address this question, we used representational similarity analysis[37], which has been used successfully in previous studies for evaluating to what degree population neural activity is categorical[19].

First, we created a continuum of pitch contours by partitioning syllables of each tone category into four groups according to either speaker-normalized pitch height or pitch change (Fig. 5g). Similar to the previous approach, the multivariate pattern

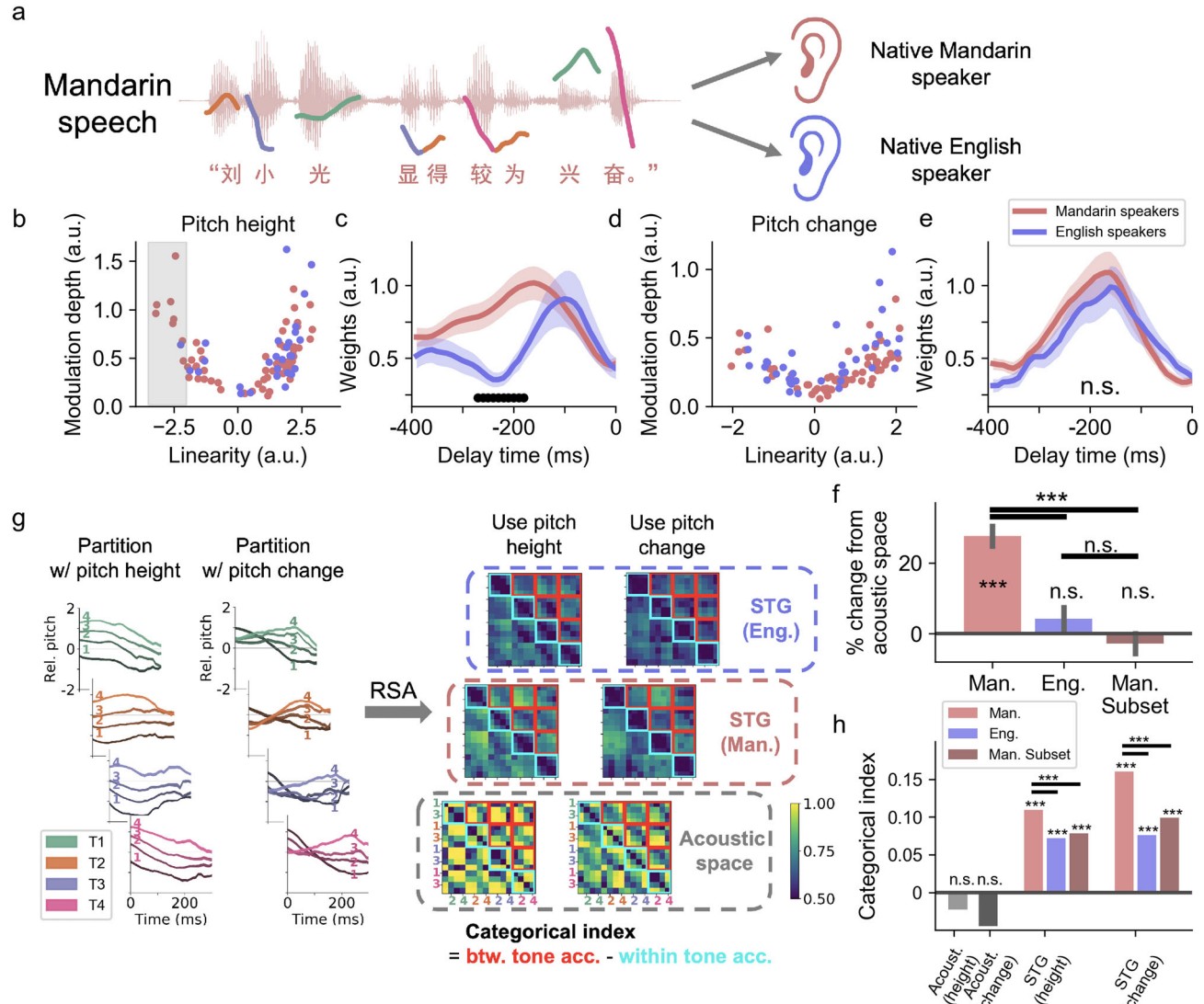

**Fig. 5 Population neural activity in human STG supports categorical representation of lexical tones shaped by language experience. a** Native Mandarin speakers and English speakers listen to the same Mandarin speech stimuli. **b** The scatterplot of the linearity versus the modulation depth for the tuning curves of relative pitch height in STG electrodes from native Mandarin speakers (red) versus native English speakers (blue). More negatively tuned electrodes are found in native Mandarin speakers (shaded gray area). **c** The averaged temporal receptive field (absolute beta weights of the encoding regression model) for relative pitch height encoding electrodes in Mandarin (red) and English speakers (blue). (mean ± s.e.m., black markers indicate time points with significant difference between the two groups, FDR corrected $\alpha < 0.05$, two-sample two-sided $t$ test). **d** Same as **b**, but for pitch change. **e** Same as **c**, but for pitch change. **f** The pair-wise tone classification accuracy relative to the accuracy in the acoustic space, using speech-sensitive electrodes (red: native Mandarin speakers; blue: native English speakers; dark red: native Mandarin speakers excluding strong negative tuning electrodes shown in **b**); error bars indicate standard error of the mean estimation from $n = 200$ times bootstrapping. **g** Using partitions based on averaged relative pitch height and averaged pitch change respectively, representational similarity analysis (RSA) is performed on acoustic space and the population STG activity in native Mandarin speakers and native English speakers, respectively. Categorical index = mean between-tone classification accuracy (red blocks) – mean within-tone classification accuracy (cyan blocks). **h** The categorical index based on acoustic relative pitch contour, STG population space in native Mandarin speakers (red) and STG subset excluding negative tuning electrodes (dark red), as well as in native English speakers (blue) (***$p < 0.005$, n.s. $p > 0.1$, permutation test).

classifier[36] was applied to single-trial neural responses and pitch contours to evaluate the representational similarity between different groups of syllables in the concatenated neural population space and the acoustic space, respectively. The critical metric we evaluated was the degree to which responses to between-tone categories and within-tone categories were similar to each other. High similarity within the tone category but low similarity between categories suggests categorical representation.

Therefore, we compared the neural confusion matrices from native Mandarin speakers and native English speakers for the 16 groups of syllables at the peak decoding time against the

confusion matrix in the acoustic space (Fig. 5g). In the acoustic space, discriminability was high for both between-tone pairs and within-tone pairs, while in the neural space discriminability was higher for between-tone pairs compared to within-tone pairs, in both Mandarin and English speakers (Fig. 5g and Supplementary Fig. 12).

To quantify the categorical structure for lexical tones, we defined the "categorical index" (CI) as the difference between the average between-tone discriminability and the average within-tone discriminability. Large positive CI indicates strong categorical representation structure. As shown in Fig. 5h, the acoustic

space did not show significant categorical structure under either continuum partitions (CI = −0.023 and −0.045, p > 0.5, permutation test), indicating that there is a comparable amount of between- and within-tone variance in the acoustic space. In the neural space, however, significant categorical structure was found in both language groups and for both continuum partitions (Mandarin speakers: CI = 0.11 and 0.16, both p < 0.005, permutation test; English speakers: CI = 0.072 and 0.076, both p < 0.005, permutation test). Therefore, tokens with the same degree of acoustic pitch difference were much easier to discriminate if they belonged to different lexical tones, which supports categorical representation of lexical tones in STG population. Specifically, the scale of the CI in native Mandarin speakers was 52% and 111% higher than native English speakers under the two continuum partitions, suggesting a strong influence of language experience at the population-level in STG. Similar to the overall tone category decoding, taking out the strong negative encoding electrodes in native Mandarin speakers resulted in significant reduction in CI (CI = 0.078 and 0.099 respectively, both p < 0.005, permutation test; Fig. 5h).

## Discussion

Pitch conveys rich lexical and intonational information and can be used for different purposes across languages. By some estimates, more than half of the world's spoken languages use linguistic pitch contrasts to distinguish word meanings[38]. These tonal languages may have vastly distinct tone systems. Therefore, how tones are represented in tonal language speakers and whether such representation is language-specific are the critical questions in speech neuroscience.

Our high-resolution cortical recordings revealed that the local cortical sites in STG encode speaker-normalized pitch features of height and change. Unlike absolute pitch, which is represented throughout the auditory system, speaker-normalized pitch is a general, but high-order auditory cue. Sensitivity to speaker-normalized pitch is not specific to Mandarin, but also used in intonational processing in English[18], and may also be implicated in other sounds such as vocal music[39,40]. Our results suggest that the neural representation of speaker-normalized pitch underlying lexical tone perception overlaps with the neural representation of intonation and is shared across languages, reflecting general auditory processing.

Pitch change and height can be independently encoded, and this has significant implications for models of pitch perception in linguistics. We found important differences in Mandarin and English speakers with regard to tuning curves and TRFs for speaker-normalized pitch features at local STG scale, possibly reflecting distinctions in the pitch statistics of the two languages. Mandarin may need wider range, especially low relative pitch below neutral pitch height, and English may need more high pitch detectors than low. These adaptive properties may account for the differences in population tone category coding in Mandarin and English speakers. As shown in Fig. 1 and Supplementary Fig. 1, successful decoding of lexical tones would require logical AND operation over time in the space of speaker-normalized pitch height and change. Therefore, we predict that, owing to the overall lack of negative pitch height sensitivity and shorter temporal integration in native English speakers, the discriminability of Tone 3 from other tones would be lower than native Mandarin speakers (Supplementary Fig. 14).

Theoretical models have traditionally posited either speech-specific or general auditory processing to the higher-order auditory cortex, with only one representation accommodated at each hierarchical processing level[41–43]. Instead, the detailed spatiotemporal information afforded by high-density cortical

recordings supports an emerging model of speech processing at multiple spatial scales: one that supports both general auditory encoding of relevant acoustic properties (e.g., speaker-normalized pitch), and also language-specific "tuning" of those properties that supports meaningful linguistic perception (e.g., lexical tones). Our results reveal both linear veridical coding of complex acoustic properties, and nonlinear, categorical representation of lexical tones at different scales. Specifically, local STG sites show strong encoding for speaker-normalized pitch but weak encoding for lexical tone categories. Population encoding across the distributed STG network show evidence for stronger categorical bias. This is consistent with previous human electrophysiological evidence that STG can support categorical perception at a population-level[19,44], and that the same sensory area can be involved in both pre-lexical and lexical processes[45]. Therefore, our results suggest that the non-primary auditory cortex can be involved in both general auditory and linguistic processing.

However, it is worth pointing out that owing to the limit in the coverage of ECoG grid, we cannot evaluate the long-range fronto–temporo–parietal connections or the potential involvement of other areas during the linguistic perception processes[13,41,46]. Therefore, it remains unclear whether the linguistically relevant categorical representation emerges from recurrent interactions within the non-primary auditory cortex or is a result of reciprocal top-down influence from other interacting areas in the language network.

Previous studies have suggested differential tone-related activation in the bilateral hemispheres, especially a leftward bias in frontal and temporal regions of native tonal language speakers and a rightward bias for non-tonal language speakers;[13] some studies also suggest a linguistic modulation effect in left temporal areas[47]. It is also suggested that speech prosody information, such as sentence-level pitch, is primarily processed in the right hemisphere[48]. Owing to the unilateral coverage in our single cases, it is hard to draw a definitive conclusion on the lateral bias in STG from our results. Compared with speech-responsive electrodes in the left STG (Supplementary Fig. 8), similar encoding of speaker-normalized pitch is observed in the right STG of native Mandarin speakers (Supplementary Fig. 9), as well as in left STG electrodes of native English speakers (Supplementary Fig. 7). Although it is not entirely clear whether these linguistic-related biases and modulations also apply to bilateral STG, our results do not preclude an effect of language experience on local STG sites. Instead, they suggest that the majority of the lexical tone-induced variance in local STG activity is driven by speaker-normalized pitch features, which are extracted in a context-dependent way.

Overall, our results suggest that Mandarin and English, as examples of tonal and non-tonal languages, share an underlying high-order auditory code for speech-related pitch in the human non-primary auditory cortex. This high-order auditory processing may be shaped by specific language experience to suit the linguistic statistics of a given language.

## Methods

The experimental protocol was approved by the Institutional Review Board at the University of California, San Francisco (UCSF) and by the Huashan Hospital Institutional Review Board of Fudan University. All participants gave their written, informed consent prior to testing.

**Participants**. This study included 15 monolingual participants (seven male, eight Female, age from 31–55, all right-handed) who were neurosurgical patients at either Huashan Hospital or UCSF. The 11 native Mandarin-speaking participants from Huashan Hospital (M1–M11) were eloquent brain tumor patients undergoing awake language mapping as part of their surgery (seven left hemisphere coverage: M1–M7, four right hemisphere coverage: M8–M11). Only the patients undergoing awake surgery with intraoperative cortical mapping were asked to participate in the study. We only included those participants with tumors that did not obviously invade the auditory cortex. Four native English-speaking participants from UCSF

(E1–E4) were patients with intractable epilepsy who had high-density electrode grids implanted for clinical monitoring of seizure activity (four left hemisphere coverage). The placements of the grids were determined solely by clinical needs. All patients were clearly informed (as detailed in the IRB-approved written consent document signed by the participants) that the participation in the scientific research was completely voluntary and would not directly impact their clinical care. Additional verbal consent was also acquired at the beginning and during the breaks of each experiment session.

**Data acquisition and neural signal processing**. The same type of high-density ECoG grids (Integra or PMT) with identical specifications (4 mm center-to-center spacing and 1.17 mm diameter exposed contact lateral) were used in both patient groups. Depending on the exact clinical need, the grid may have 128 (16 × 8) or 256 (16 × 16) contact channels in total. During experimental tasks, neural signals were recorded from the ECoG grids using a multichannel amplifier optically connected to a digital signal processor (Tucker-Davis Technologies). The TDT OpenEx software was used for data recording. The local field potential at each electrode contact was amplified and sampled at 3052 Hz. The raw voltage waveform was visually examined, and channels containing signal variation too low to be detectable from noise or continuous epileptiform activity were removed. Time segments on remaining channels that contained electrical or movement-related artifacts were manually marked and excluded. The signal was then notch-filtered to remove line noise (at 50 Hz, 100 Hz, and 150 Hz for Mandarin-speaking participants and at 60 Hz, 120 Hz, and 180 Hz for English-speaking participants) and re-referenced to the common average across channels sharing the same connector to the preamplifier.

Using the Hilbert transform, the analytic amplitudes of eight Gaussian filters (center frequencies: 70–150 Hz) were computed. The high-gamma signal was taken as the average analytic amplitude across these eight bands. The signal was downsampled to 100 Hz. The tasks were broken into recording blocks of ~5 min in length. The high-gamma signal was z scored across each recording block.

**Experimental stimuli**. The acoustic stimuli used in this study consisted of natural, continuous speech in both Mandarin Chinese and American English. The Mandarin speech was a subset of the Annotated Speech Corpus of Chinese Discourse (ASCCD) from the Chinese Linguistic Data Consortium (http://www.chineseldc.org), which included reading texts of a variety of discourse structures, such as narrative and prose[20]. The stimuli set consisted of 68 passages of Mandarin speech selected from the ASCCD corpus, spoken by 10 different speakers (five males, five females). The length of single passage varied between 10 and 60 s. Across all passages, we had a total of 4711 tokens (syllables) of the four lexical tones. The passages were separated with 0.5 s of silence. The task was broken into six blocks with each block ~5 min in time.

The English speech stimuli consisted of materials from the TIMIT corpus[21] and the Boston University Radio Speech Corpus (BURSC)[22]. The TIMIT set consisted of 499 English sentences selected from the TIMIT corpus, spoken by 402 different speakers (286 males and 116 females). The sentences were separated with 0.4 s of silence. The task was broken into five blocks with each block ~5 min in time. The BURSC set consisted of 75 passages of English speech selected from the BURSC corpus, spoken by six different speakers (three males, three females). The length of single passage varied between 10 and 60 s. The passages were separated with 0.7 s of silence. The task was broken into six blocks with each block ~5 min in time.

Depending on their clinical conditions, each participant finished between 6 and 17 blocks of all tasks. In particular, all 15 participants finished at least four Mandarin ASCCD blocks. Four English-speaking participants (E1–E4) and four Mandarin-speaking participants (M2–M5) also finished at least three English blocks. See Supplementary Table 1 for a detailed list of tasks finished by each participant.

**Data analysis software**. All analyses were carried out using custom software written in MATLAB, Python and R. Open-source scientific Python packages used included numpy, scipy, pandas, and scikit-learn. Open-source R package standGL[49] was used. Cortical surface reconstruction was performed using Freesurfer and electrodes were co-registered using Python package img-pipe. Praat[50] was used to extract pitch features. Figures were created with matplotlib and seaborn in Python.

**Spectrum and pitch feature extraction (Fig. 1)**. The spectrum features of the speech were calculated using a mel-band auditory filterbank of 30 filters ranging from 0 to 8KHz. Absolute pitch was calculated using procedures identical to Tang et al.[18], where the fundamental frequency ($F0$) was calculated using an automated autocorrelation method in Praat and corrected for halving and doubling errors. The absolute pitch was defined as the natural logarithm of $F0$ values in Hz. The relative pitch was computed by $z$ scoring the absolute pitch values (log$F0$) within each sentence/passage (within speaker). The pitch change was computed by taking the first-order derivative (finite difference) in time for log$F0$. We discretized absolute pitch, relative pitch, and pitch change into 10 bins, equally spaced from the 2.5 percentile to the 97.5 percentile value. The bottom and top 2.5% of the values were placed into the bottom and top bins, respectively. As a result, absolute pitch, relative pitch, and pitch change were represented as three 10-dimensional binary feature vectors. For non-

pitch periods, these feature vectors would have all 0 s. See Supplementary Fig. 4 for a demonstration of feature extraction.

**Principal component analysis (PCA) of lexical tones**. We used PCA to analyze the acoustic pitch space of lexical tones. We extracted the pitch contours of the 4711 tone exemplars in the ASCCD corpus. These contours were time-warped to have the same length of 250 ms (25 time points at 100 Hz sampling rate). PCA was performed on this $4711 \times 25$ data matrix $X$, and we got decomposition $X = LW^T$, where $L$ is a $4711 \times 25$ PC score matrix, and $W$ is a $25 \times 25$ orthogonal weight matrix with columns of $W$ forming an orthogonal basis set for the 25 temporal features.

**Electrode localization (Fig. 2)**. For the chronic monitoring cases, electrodes were localized by aligning preimplantation MRI and post-implantation CT scans. For the awake cases, high-density electrode grids were temporarily placed onto the superior temporal gyrus intraoperatively to record cortical local potentials[19]. The three-dimensional positions of the corners of the grid were recorded using the Medtronic neuronavigation system and then aligned to the pre-surgery MRI[51]. Intraoperative photographs were used as references. The remaining electrodes were localized using interpolation and extrapolation from those points.

**Speech-responsive electrodes (Fig. 2)**. To find speech-responsive electrodes, we first aligned high-gamma responses to the onsets of speech. Onsets were defined as times in the stimulus where sound was preceded by at least 400 ms of silence. We then used a paired sample $t$ test to test whether the average response after speech onset (window from 100 ms after onset to 400 ms after onset, accounting for a neural delay of ~100 ms) was significantly different from the response before speech onset ($-250$–$50$ ms) ($p < 0.01$, $t$ test, paired two-sided, Bonferroni corrected for total number of electrodes). Because the temporal dynamics of the response differ between electrodes, we also tested neural responses at speech offsets, comparing average high-gamma before speech offset to average high-gamma after speech offset. This comparison captured electrodes that did not have a strong onset response, but whose response was elevated before speech offset and dropped off after speech offset. Offsets were defined as times in the stimulus where the sound was followed by at least 400 ms of silence. All $t$ tests were corrected for multiple comparisons using the Bonferroni method.

**Tone discriminant electrodes (Fig. 2)**. To find speech-responsive electrodes that also discriminate between lexical tones, we first aligned high-gamma responses to the onsets of the tones. We then used $F$ test to test whether the mean high-gamma responses of the four tones were significantly different. Specifically, we compute the $F$ statistic for every time point during the $-200$ ms to 1000 ms time period relative to the onset (120 total time points) and find significant time points with two-sided $p < 0.05$ threshold using Bonferroni correction for total number of electrodes and time points. Only the electrodes with at least three consecutive significant time points were considered as tone discriminant electrodes.

**Pitch temporal receptive field (TRF) analysis (Figs. 2–3)**. To determine whether speaker-normalized pitch features drive neural responses in lexical tone-discriminating neural populations, we used time-delayed linear encoding models known as TRF models[52]. TRF models allow us to predict neural activity based on stimulus features in a window of time preceding neural activity. In particular, we fit the linear model $y(t) = \sum_{f=1}^{F} \sum_{\tau=0}^{T} \beta_f^T(\tau) x_f(t-\tau) + \epsilon$ for each electrode, where $y$ is the high-gamma activity recorded from the electrode, $x_f(t-\tau)$ is the stimulus representation vector of feature set $f$ at time $t-\tau$, $\beta_f(\tau)$ is the regression weights for feature set $f$ at time lag $\tau$, and $\epsilon$ is the gaussian noise.

In the full TRF model, we included features for the sound spectrum, sound intensity, absolute pitch, relative pitch, pitch change, and tone category. (Note that tone category features are only applicable for Mandarin speech corpus. When comparing the Mandarin model and the English model, we only used the common features shared by the two languages. See next section for details.) To calculate the unique contributions of specific features, such as speaker-normalized pitch features (relative pitch and pitch change), we fitted TRF models that excluded these features and calculated the difference in $R^2$ between the full and reduced models.

To prevent model overfitting, we used L2-norm regularization and cross-validation. Specifically, we divided the data into three mutually exclusive sets of 80%, 10%, and 10% of samples. The first set of 80% was used as the training set. The second set was used to optimize the L2 regularization hyperparameter, and the final set was used as the test set. We evaluated the models using the correlation between actual and predicted values of neural activity on held out testing data. We performed this procedure five times and the performance of the model was taken as the mean of performance across all test sets.

To calculate the significance of unique portions of variance explained, we employed permutation testing. When performing the permutation test, we would like to keep the intrinsic temporal structure in the speech stimuli while permuting the correspondence between the stimuli and neural activity. Therefore, we found phrases in the continuous speech and shuffled at the level of phrases. Here a phrase

is defined as a continuous speech segment with at least 150 ms of silence before and after. We shuffled the acoustic features between all phrases in the stimuli before computing null values of the unique variance explained by absolute pitch, by tone category, or by speaker-normalized pitch features by running the same analysis pipeline. We ran this procedure 200 times to get a null distribution of values. Unique $R^2$ values above the 99th percentile were considered significant.

To compare the representation of speaker-normalized pitch and absolute pitch (or tone category), we used paired $t$ test to compare the unique $R^2$ for speaker-normalized pitch against the unique $R^2$ absolute pitch (or tone category) on electrodes that were significant for either speaker-normalized pitch or absolute pitch (tone category), including those that were significant to both.

All speech-responsive electrodes in Mandarin (M1–M11) and English (E1–E4) speakers were included in the TRF analysis.

**Comparison of Mandarin-fit and English-fit models (Fig. 4).** To determine whether the neural response to Mandarin tones could be predicted from neural responses to English speech, we recorded cortical activity while participants listened to English speech (English data), in addition to the cortical activity recorded while participants listened to Mandarin (Mandarin data). We constructed the stimulus representations for both the English and Mandarin speech stimuli using the same feature descriptions. These features included the acoustic spectrogram, intensity, absolute pitch, relative pitch, and pitch change. We then fitted TRF models on the English data and used the regression weights to predict neural responses to Mandarin speech.

To quantify the performance of the Mandarin and English models, we calculated the correlation between the predicted and actual neural responses. To compare the Mandarin and English model performances, we used a bootstrapping procedure to derive a distribution of correlation values by sampling from the trials of individual tones. Specifically, we randomly chose 500 trials with replacement, calculated the correlation between the actual and predicted response to those trials, and then performed that procedure 200 times to arrive at two distributions, one for the Mandarin model and one for the English model. We tested whether those distributions were significantly different using the Kolmogorov–Smirnov test.

To determine whether speaker-normalized pitch encoding was similar for individual electrodes between Mandarin and English, we calculated the correlation of the regression weights of the two models. To reduce noise in the weights, we first spatially blurred the 2D weight matrices (no. of time lags by no. of feature dimensions) using a 2D-gaussian filter with sigma of 1 (the filter was applied to each feature group separately), and then calculated the correlation. Positive correlations indicated that an electrode had similar tuning for speaker-normalized pitch features across the two languages.

All subjects that finished both ASCCD blocks and at least one block of each of the two English tasks (M2–M5, E1–E3) were included in this analysis.

**Tuning curve and TRF (Fig. 5).** To evaluate the functional relationship between the speaker-normalized pitch features and the neural activity in STG, we computed the tuning curve for each speech-responsive STG electrode. The relative pitch height and pitch change were discretized into 20 bins uniformly spanning the middle 95-percentile range. The mean and standard deviation of high-gamma activity in each electrode was calculated for each bin. The modulation depth was defined as the absolute difference between the maximum and minimum mean high gamma. The linear correlation coefficient ($r$) between the pitch features and mean high gamma in the tuning curve was estimated. The linearity ($L$) was defined as the Fisher transformation of the slope, $L = \frac{1}{2}\ln(\frac{1+r}{1-r})$. The TRF of each electrode was computed as the absolute average beta weights at each time point.

**Population tone decoding (Fig. 5).** To determine whether neural responses to different tones were distinguishable and to evaluate the representation structure of lexical tones in the speech-responsive STG network, we aligned high-gamma responses to the onsets of the vowels of syllables and divided trials by the lexical tone of the syllable. A sliding time window with a length of 50 ms (five consecutive time points of high-gamma activity) was used to evaluate the dynamics of neural representation. For each sliding window, the neural activity across the five consecutive time points in all speech-responsive electrodes were concatenated and used as features to train a pattern classifier. Considering the relatively small number of tone-encoding electrodes and the inconsistency in grid coverage in single patients, we pooled speech-responsive electrodes in all Mandarin-speaking participants to construct the neural feature space. We first computed the average pair-wise classification accuracy across all six pairs for each sliding time window, and then took the maximum accuracy across all sliding windows as the "peak overall pair-wise classification accuracy". For the acoustic space, we took the entire time course of 250 ms as input features (25 time points in total), and at each time point both relative pitch height and pitch change were used.

Specifically, we used logistic regression with group-lasso penalty across electrodes[36,49], where L2-norms of all the temporal coefficients from each electrode were summed to induce an L1-norm penalty across electrodes. This approach would avoid overfitting while maintaining temporal smoothness within electrode and promoting sparse interactions between electrodes. A nested cross-validation strategy was adopted where five-fold cross-validation was used to estimate the

classification accuracy, and within each training set, ten-fold cross-validation was used to select the optimal penalty parameter $\lambda$.

For the between-tone classification, we computed the averaged pair-wise accuracy (six pairs in total) for both the acoustic space ($Q_{aco}$) and neural population at each time point for Mandarin-speaking and English-speaking subjects ($Q_{man}(t)$ and $Q_{eng}(t)$). We used the accuracy over chance level (Q-0.5) as the metric of classification performance and used the performance in the acoustic space as baseline. Finally, we reported the maximum percent of change in the classification performance for the speech-responsive STG population in Mandarin-speaking and English-speaking groups over time ($t$), relative to the baseline in acoustic space.

**Representational similarity analysis (Fig. 5).** For the representational similarity analysis, syllables were grouped into 16 groups, where all the syllables within each lexical tone were sorted into four groups according to the relative pitch height (PC1) or pitch change (PC2). The representation dissimilarity matrix (RDM) was constructed using pair-wise classification accuracy between all possible pairs of syllable groups at the peak time based on the overall between-tone classification accuracy.

Permutation test was used to determine the statistical significance of the results. For each permutation, the group labels of all the syllables were randomly shuffled, and the exact same procedure was performed to compute the RDM for each sliding time window. The permutation was repeated 200 times, and the accuracy and category index were computed for each permutation. Any value above the 95th percentile were considered as significant.

To maximize the statistical power, we only included speech-responsive electrodes in subjects that finished all ASCCD blocks (M1–M7 and E1–E3).

**Reporting summary.** Further information on research design is available in the Nature Research Reporting Summary linked to this article.

## Data availability
The data set generated during the current study will be made available from the authors upon reasonable request. Source data are provided with this paper.

## Code availability
The completely developed code that operates on the full data set will be made available from the authors upon reasonable request.

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

## Acknowledgements

We thank Yulia Oganian, Shelley Tong, Ilina Bhaya-Grossman, Matthew Leonard, Keith Johnson, and Bharath Chandrasekaran for critically reading the manuscript and for helpful suggestions. We would also like to thank Yanming Zhu for the help with data collection. The authors gratefully acknowledge the support of the National Institute of Neurological Disorders and Stroke under U01NS117765 (to E.F.C.), the National Institute on Deafness and Other Communication Disorders under R01DC012379 (to E.F.C.), the William K. Bowes Foundation (to E.F.C.), the William and Susan Oberndorf Foundation (to E.F.C), the Joan and Sanford Weill Foundation (to E.F.C.), the Shurl and Kay Curci Foundation (to E.F.C.), Shanghai Municipal Science and Technology Major Project under No. 2018SHZDZX01 (to J.W.), Shanghai Shenkang Hospital Development Center under SHDC12018114 (to J.W.), Shanghai Rising-Star Program under No. 19QA1401700 (to J.L.) and Shanghai Young Talents Program under No. 2017YQ014 (to J.L.).

## Author contributions

Conceptualization, E.F.C., C.T., and Y.L.; investigation, Y.L., C.T., and J.L.; formal analysis, Y.L., C.T., and J.L.; writing—original draft, Y.L. and C.T.; writing—review & editing, Y.L., J.L., J.W., and E.F.C; resources: J.W. and E.F.C.; supervision: J.W. and E.F.C.

## Competing interests

The authors declare no competing interests.

## Additional information

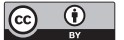

