## [Peer Review File · Nature Communications]

REVIEWER COMMENTS

Reviewer #1 (Remarks to the Author):

Li et al. studied 11 native Mandarin Chinese speakers with brain tumor who underwent intraoperative intracranial recording. They also studied four native English speakers with drug-resistant epilepsy who underwent intracranial EEG (iEEG) recording at the bedside. During the intracranial recording, the study participants experienced auditory tasks, in which they were instructed to listen to Mandarin and English speech stimuli (spoken sentences). The authors determined how the tone of speech stimuli would be processed in the superior temporal gyrus (STG) of patients who speak Mandarin (tonal language) and English (non-tonal language). They analyzed whether specific pitch features would be associated with corresponding high-gamma (70-150 Hz) activity patterns in the STG. The authors employed machine learning at each STG electrode in Mandarin speakers to learn the relationship between English tonal features and high-gamma activity patterns. The learned relationship could predict the STG high-gamma patterns related to Mandarin sounds in the same patient group. It is unclear whether the relationship between Mandarin tonal features and high-gamma power patterns was likewise capable of predicting the high-gamma power patterns related to English sounds. It is unclear whether the same analysis was systematically performed in English-speaking patients. Based on the machine learning-based classification, as employed on all STG electrodes available on each patient, Mandarin pitch features were classified by the STG high-gamma activity of Mandarin speakers better than English speakers. It is unclear whether a similar analysis was performed to determine whether pitch features in English were classified by English speakers than Mandarin speakers. They conclude that speech perception relies upon a shared auditory feature processing mechanism in the STG and that this processing is tuned to a given language.

(1) The study provided novel observations that, at least in part, support their conclusion. The strength of this study also includes state-of-art electrophysiology and machine learning analyses.

(2) Results: We found some of the electrodes appear to be located outside the STG, though the authors claim that they analyzed STG electrodes.

(3) Abstract and main text: Though the abstract infers that analysis was systematically performed in two distinct patient groups (i.e., Mandarin/intraoperative recording vs. English/bedside recording), we found that some analysis was performed only in one of the two groups (or the results were described for only one group). Thus, we are not sure whether the results of this study fully support all of the conclusions.

(4) Methods/Discussion: It is unclear how they controlled the effects of different experimental conditions on high-gamma activity. Mandarin speakers were given auditory stimuli during the awake craniotomy, whereas English speakers underwent the task at the bedside. Mandarin speakers had brain tumors, whereas English speakers had drug-resistant epilepsy. Some Mandarin speakers had electrodes placed on the right hemisphere, whereas English speakers had electrodes exclusively on the left hemisphere.

(5) Methods/Discussion: It is unclear whether the study included bilingual patients. This information may be useful to determine whether the high-gamma patterns are indeed tuned to a given language.

(6) Methods: The authors should provide the specifications of high-density electrodes used for each patient group.

(7) Methods/Supplementary document: The authors should describe the specifications (e.g., number of syllables) of Mandarin and English auditory stimuli.

(8) Methods: The authors should define "specific recording block" on Line 510.

(9) Methods: The authors should explicitly describe the electrode montage/reference used for intracranial recording.

(10) Methods: The authors should provide the number of speech responsive STG electrodes in 'pitch temporal receptive field analysis' and 'population tone decoding and representational similarity analysis.'

Reviewer #2 (Remarks to the Author):

This study explores the auditory and linguistic representation of Mandarin lexical tones in superior temporal gyrus using ECoG. The data show strong and convincing evidence for *auditory* encoding of lexical tones based on speaker-normalized relative pitch height and pitch change. Evidence for *linguistic* processing of lexical tones in STG is sought (and found) in the comparison of Mandarin vs. English speakers' neural responses to Mandarin speech. Results converge across a number of cutting-edge analyses of high gamma activity, including modelling of single electrode data with temporal response functions (TRFs) within Mandarin and across Mandarin and English speech, as well as multivariate pattern classification and representational similarity analyses at population level.

To my knowledge, this is to date the most compelling study on the encoding of lexical tone, and it integrates well into previous work of the same lab on categorical phoneme perception (Chang et al., 2010, Nat Neurosci) and intonation perception (Tang et al., 2017, Science) in non-tonal languages. Results are novel, will influence thinking in the field, and are also of interest to a broader readership, given that about half of the world's languages are tonal. Authors succeed in providing within-study replication of results by complementary analyses, all results are corrected for multiple comparisons.

Still, I do have a few major points that the authors may wish to address in a revision to allow other researchers to reproduce their work and to better link the results to prevailing language models. More precisely, the choice of acoustic features (e.g., the exclusion of spectrum), their extraction from the speech signal, and the construction of TRF models need better motivation / specification (major points [1] to [3]). Moreover, given that Mandarin (but not English speakers) comprehend Mandarin speech, it should be discussed whether STG truly encodes linguistic tone categories, or whether results reflect the top-down modulation of auditory processes by remote linguistic processes (major points [4] and [5]).

Major points:

[1] Extraction of acoustic features (l. 83-100). The methods leading to conclude that "relative pitch height and pitch change are speaker-normalized acoustic features that define lexical tones in Mandarin" (l. 83-84) are hardly presented in the Methods section. Please add a description to the Methods of how relative pitch height and pitch change were calculated (as shown in Figure 1). Also the PCA, its execution and results need more explanation. What did you enter into the analysis? Did you calculate one PCA for each time point? Does "average" in S1C and S1D mean "average across exemplars from each tone"? Did you use Praat for pitch extraction? How many tone exemplars were entered into the analyses, etc.?

[2] Choice of acoustic features – exclusion of spectrum (l. 102-187). The Methods section should specify how / with which software spectrum and intensity were extracted and how TRF models were built. What does "spectrum" refer to – the spectral centroid, SD of spectrum or else? Judging from the unique R2 values (lines 147-148), spectrum actually looks like a very good predictor – even better than speaker-normalized features. How would spectrum compare to speaker-

normalized features in Fig. 3, and what does it mean for the conclusion that “the differential neural responses with regard to lexical tones are ‘mainly’ driven by the encoding of speaker-normalized pitch features” (lines 183-184)?

[3] TRF models (l. 139-187). How exactly were TRF models built, e.g., how were non-pitch periods (consonants, pauses etc.) coded? A supplementary figure would help (e.g., like Fig. 1A in Di Liberto et al., 2020, eLIFE), along with a more detailed description in the Methods section.

[4] Interpretation of neural tuning shaped by language experience (l. 300-338). I find it surprising to see that integration windows were *shorter* in English than Mandarin listeners (Figure 5C). Given that pitch is a suprasegmental feature in non-tonal languages, typically spanning several syllables, wouldn't one expect to see *longer* integration windows in English than Mandarin listeners? Hence, are these integration differences driven by “statistical differences” between English and Mandarin, or by the fact that Mandarin (but not English) speakers comprehend the presented material, triggering top-down linguistic processes? Please add a comment on that in the manuscript. [One way to experimentally address this question may be to investigate whether integration windows in English listeners are longer when listening to their native English language than to Mandarin speech. Likewise, to dissociate the influence of speech statistics and top-down comprehension related processes, it may be interesting to add another language with similar statistics, but that listeners don't understand (e.g., German or Thai or tonal chimeras/jabberwocky) in future work.]

[5] Discussion. “Therefore, our results provide evidence that STG is involved in both general auditory and *linguistic processing*.” (l. 451-452); “Speech perception relies upon a shared auditory feature processing mechanism in the cortex, which is tuned to the *statistics of a given language*.” (abstract). These conclusions / wordings seem too strong. The fact that neural activity discriminated tone categories to a significant degree also in English native speakers indicates that processes are primarily auditory rather than linguistic (Figure S5, purple bars in Figure 5H), as acknowledged by the authors. An explanation that should be considered is the influence of top-down linguistic processes due to language experience/comprehension that drove/shaped auditory processes in STG via long-range fronto-temporo-parietal connections in Mandarin speakers (see also comment [3]).

Minor points:

- General: It would be helpful if the authors indicated in the Methods which analyses correspond to which Figure(s) or Figure panels.
- General: Methods switch between past and present tense; tense could be more consistent.
- L. 479-494: Please add a note on whether Mandarin (English) participants were able to comprehend the English (Mandarin) material or not.
- L. 556: typo – change “discriminant” to “discriminate”.
- L. 554-562: Please specify whether all electrodes or only speech responsive electrodes were inspected in this analysis?
- L. 118: typo – change “Fig. 2(B, H)” to “Fig. 2(B, G)”.
- L. 587-592: “We shuffled the acoustic features between all phrases ...” – what is meant by phrases?
- L. 164-175: Please specify in the Methods section that t-tests were run only on unique R2 values of electrodes that were responsive either to speaker-normalized pitch OR absolute pitch (tone category). Those that responded to both were excluded, weren't they?
- L. 180-181: Please specify what “F-statistics across time” means. Mean or median F-value across time, and in which time-window relative to tone onset?
- L. 234: typo – change “Figure 4A” to “Figure 4B”.
- L. 240: typo – change “(Fig. S3G, H)” to “(Fig. S4G)”.
- L. 248-252: “To quantify this similarity across all neural populations ...” and “For nearly all electrodes ...” – does this mean all electrodes, all speech-responsive or all tone-responsive electrodes?

- L. 616: "we first spatially blurred the weights" – how was this done?
- L. 272-275: "Mandarin/English group" and "within Mandarin group" sounds like referring to participant groups (Mandarin and English native speakers). Please rephrase.
- Figure S4 G-I: Given that "Mandarin tone contours have a larger dynamic range of pitch and are temporally organized by syllable-sized segments compared to English speech" (lines 302-303), it seems rather natural/expected that the Mandarin model outperforms the English model. What is it that the authors intend to say with these comparisons?
- L. 623: Does "speech-responsive STG electrodes" mean that speech-responsive electrodes outside of STG were excluded?
- L. 636 and 640: "in the STG network" vs. "all speech responsive electrodes" – please specify because not all speech responsive electrodes were located in STG.
- L. 640: "neural activity across all speech responsive electrodes" – please specify that/whether "neural activity" refers to mean HG in the 50ms window, per electrode.
- L. 352: What does "peak" in "peak overall pairwise classification accuracy" refer to? The sliding window, i.e. time? Or tone pair? Both?
- L. 660 and legend of Figure S5: What is "STG space"? More generally, the second half of the manuscript talks about STG activity patterns, STG space and similar, i.e., an anatomical choice of electrodes in contrast to the choice based on functional speech-responsiveness at the beginning of the manuscript.
- Figure S9 needs more explanation. What does it show? How was it obtained? How does it show that tone 3 is less discriminable from the other tones in English than Mandarin native speakers?
- L. 466: "which is extracted in a context-dependent way" – please specify what "context" means.

Reviewer #3 (Remarks to the Author):

In this manuscript from Li et al, the authors investigate how lexical tones present in Mandarin speech are represented in human non-primary auditory cortex, using high-density ECoG recordings. They find that high gamma-activity in individual electrodes can encode features relevant for discriminating between lexical tones (speaker normalized pitch height and change). Using models based on English rather than Mandarin produced similar regression weights (positive correlation) across electrodes, suggesting speaker-normalized pitch encoding was a general feature of non-primary auditory cortex, and did not require a tonal language. Furthermore, native English speakers also could represent pitch height and change, like Mandarin speakers. But while pitch change showed similar neural tuning, pitch height tuning had a greater dynamic range in native Mandarin speakers (compared to native English speakers), likely due to the different use of pitch in these two languages.

The experiments are well designed and thoughtfully analysed, and the results are an important step forward for the field. I only have a few minor comments aimed to improve the clarity and overall understanding of the results.

Minor comments:

1. In fig 2B/G- actual HG zscore never goes above 1, even though the activity is significant different between tones. I assume this is due to the z-score being calculated over the entire recording block. If instead you calculate the z-score using the mean and standard deviation of the recording period when no sound was present (which presumably would have less neural activity), would the zscore be significantly different from baseline? In other words, are the neurons driven by the stimulus?
2. Fig 2E (complete model) looks much more like the real data, but is due to greater complexity of the model. If you remove parameters such as spectrum, intensity and absolute pitch, does it still match? Can you resolve whether a better fit is due to just greater complexity or using the correct parameters.

3. Line 121- "This shows that adjacent STG electrodes may be differentially tuned to lexical tones." Are you suggesting that there is a topographical organization for lexical tone in the STG? Or is there another explanation for this result? If the neurons that underlie the recorded gamma signal are heterogenous in how they respond to lexical tones (e.g. half the neurons prefer T4>T1 and the other half prefer T1>T4), is this detectable with current methods.

4. Generally, some more raw data would help. It's good that you show the actual HG plots in Fig 2B and G, and where these electrodes were located (in A) but it would be helpful to see how this looks across all the electrodes (and highlight the specific time-amplitude plots that are significant and non-significant).

5. Line 168- Speaker normalized pitch features explained up to 6% variance. Is this a lot? You write on line 183- "Therefore, at the level of individual electrodes, the differential neural responses with regard to lexical tones are mainly driven by the encoding of speaker-normalized pitch features." This seems to contradict the 6%...

referee- Daniel Bendor

Reviewer #4 (Remarks to the Author):

The authors study the pitch encoding properties of the superior temporal gyrus (STG), as a key non-primary auditory cortical area, across native Mandarin and English speakers using electrocorticography (ECoG). They consider two main speaker-normalized pitch features, namely relative pitch height and pitch change, and demonstrate the differential encoding of these features in STG across the two speaker populations.

The first finding of the study is that in Mandarin speakers, tone category, spectrum, intensity and absolute pitch are not differentially encoded across the 4 tones in Mandarin, whereas relative pitch height and pitch change are. The second finding is that the encoding of relative pitch height and pitch change actually contribute to the discriminability of the tone categories (which is not the case for absolute pitch). Third, it is shown that the encoding models of these speaker-normalized pitch features are remarkably similar across English and Mandarin speech, suggesting a universal neural representation of these features. Fourth, using temporal receptive field (TRF) analysis, the temporal profile of the neural responses of native Mandarin and English speakers to Mandarin speech are shown to pertain to different time-scales, perhaps tuned to the language-specific statistics of tone contours. Finally, it is shown that while these speaker-normalized features are encoded at the electrode level (but not tone categories), there is a population-level emergence of categorical representation of the tones in STG.

The paper is well-written overall, the methods are compelling, and the results are significant, as they provide new evidence on the role of STG in concurrent processing of both universal and language-specific pitch features. I have the following comments that would like the authors to address:

Major comments:

1) The emergence of the population encoding of tone categories is an impressive finding. It would be helpful to visualize the weights of the multi-variate classifier (logistic regression in this case), and see if the spatial/spatiotemporal nature of these coefficients can provide further anatomical insights on the underlying population neural code across STG. For instance, is the population code the result of positive posterior and negative anterior weights in the STG electrodes? I understand

that the pitch-selective electrodes are pooled across subjects to increase statistical power, and this may hinder finding a clear anatomical structure in the regression coefficients, but probing the anatomical structure of the classifier weights would be helpful to shed light into the nature of the population code.

2) On a related note, lines 641-642 mention that the pitch-encoding electrodes were pooled across subjects to construct the feature space, which is totally fine. But, in reading pages 13-14 (regarding Fig. 5F and 5G), I was under the impression that the categorical indices were computed at the individual subject level and then averaged to get panels F and H. I suggest that the authors make this explicit on page 13 that the results follow pooling all the pitch-encoding electrodes across subjects.

3) Fig. 5, Panel H suggests that the categorical index based on the pitch change partition is higher than that corresponding to relative pitch height partition for Mandarin speakers (which is not the case for English speakers). This suggests that relative pitch height has a smaller role in the population-level categorical encoding of tones compared to the pitch change for Mandarin speakers. But, at the same time panel E shows very similar TRFs for the pitch change across English and Mandarin speakers, whereas the TRFs for the relative pitch height in panel C are quite distinct. Judging by panels C and E, one would expect that the categorical index for the relative pitch height partition would also be higher than that of pitch change (which is the opposite of what panel H shows). Do the authors have any thoughts on/explanations for this seeming discrepancy?

4) The description of group-lasso on Page 19, lines 647-650 is not accurate. I believe the authors are referring to grouping the coefficients of each electrode in the logistic regression by the L2-norm and summing the L2-norms to induce an L1-norm penalty across electrodes. If this is the case, I suggest the following rewording: "...where L2-norms of all the temporal coefficients from each electrode were summed to induce an L1-norm penalty across electrodes". Also, the statement "network interactions between local populations" is not accurate. I suggest the following rewording: "...while maintaining temporal smoothness within electrode and promoting sparse interactions between electrodes".

5) I noticed that several references to figure panels of the main text and supplementary material were mis-numbered:

- Page 4, Line 118: Fig. 2(B,H) -> Fig. 2(B,G)

- Page 9, line 234: Fig. 4A -> Fig. 4B

- Page 9, line 240: Fig. S3G,H -> Fig. S4G,H,I.

- Page 13, line 359: Fig. S6 -> Fig. S5

Minor comments:

6) Page 4, lines 119-121: the statement that the first electrode responded significantly to tone 1, 2 and 3 is not accurate. It seems that the electrode responds to all tones significantly, if significance is assessed based on the pre-onset response. Perhaps you can just say the first electrode shows higher responses to tones 1, 2, and 3.

7) Some typographical/wording suggestions:

- Abstract: auditory speech cortex -> auditory cortex

- Page 4, lines 93-94: speaker-normalized pitch height -> speaker-normalized relative pitch height

- Page 9, line 234: English-data -> English data

- Page 12, line 331: $p < 0.05$ -> $\alpha < 0.05$ (I assume 0.05 is the target FDR level, as in the caption of Fig. 5C).

REVIEWER COMMENTS

Reviewer #1 (Remarks to the Author):

Li et al. studied 11 native Mandarin Chinese speakers with brain tumor who underwent intraoperative intracranial recording. They also studied four native English speakers with drug-resistant epilepsy who underwent intracranial EEG (iEEG) recording at the bedside. During the intracranial recording, the study participants experienced auditory tasks, in which they were instructed to listen to Mandarin and English speech stimuli (spoken sentences). The authors determined how the tone of speech stimuli would be processed in the superior temporal gyrus (STG) of patients who speak Mandarin (tonal language) and English (non-tonal language). They analyzed whether specific pitch features would be associated with corresponding high-gamma (70-150 Hz) activity patterns in the STG. The authors employed machine learning at each STG electrode in Mandarin speakers to learn the relationship between English tonal features and high-gamma activity patterns. The learned relationship could predict the STG high-gamma patterns related to Mandarin sounds in the same patient group. It is unclear whether the relationship between Mandarin tonal features and high-gamma power patterns was likewise capable of predicting the high-gamma power patterns related to English sounds. It is unclear whether the same analysis was systematically performed in English-speaking patients. Based on the machine learning-based classification, as employed on all STG electrodes available on each patient, Mandarin pitch features were classified by the STG high-gamma activity of Mandarin speakers better than English speakers. It is unclear whether a similar analysis was performed to determine whether pitch features in English were classified by English speakers than Mandarin speakers. They conclude that speech perception relies upon a shared auditory feature processing mechanism in the STG and that this processing is tuned to a given language.

We thank the reviewer for the comments and feel encouraged that the reviewer appreciates the strength of this manuscript. We also thank the reviewer for bringing up questions about the cross-language comparisons in the comments. We feel that addressing these comments improves the clarity of the manuscript.

In particular, regarding to the reviewer's general comments above: "It is unclear whether the relationship between Mandarin tonal features and high-gamma power patterns was likewise capable of predicting the high-gamma power patterns related to English sounds. It is unclear whether the same analysis was systematically performed in English-speaking patients." We now show in Fig. S11 that we can use encoding model trained on Mandarin speech to predict neural response to English speech in both Mandarin- and English-speaking groups. We have also made it clear that all analyses were performed in both groups when applicable.

Regarding to the reviewer's general comment that "It is unclear whether a similar analysis was performed to determine whether pitch features in English were classified by English speakers than Mandarin speakers", we want to emphasize that pitch

representation in non-tonal language (e.g. English) has been studied in previous literature (Tang et al., Science 2017) and is not the main focus of this paper. Pitch mainly represents intonational information in English speech and there is no English counterpart of lexical tones, so we did not study the categorical representation of pitch in English speech.

Below we list our point-to-point responses addressing concerns raised by the reviewer, in particular the concerns regarding the cross-language comparisons between Mandarin- and English-speaking groups.

(1) The study provided novel observations that, at least in part, support their conclusion. The strength of this study also includes state-of-art electrophysiology and machine learning analyses.

We thank the reviewer for the positive comment on the novelty and strength of our manuscript.

(2) Results: We found some of the electrodes appear to be located outside the STG, though the authors claim that they analyzed STG electrodes.

We found an error during the electrode localization, which led to mislabeled rows in the grids for two participants (M9 and M11). After correcting the error, the speech-responsive and tone-discriminant electrodes in these two participants were all located within STG (see new Fig S3). We apologize for making that error in the original version of the manuscript.

The recordings for all the mandarin speakers were performed during awake surgeries and postoperative CT scan was not available. The localization of these electrodes relies on intraoperative photographs taken during the surgery and manual alignment. Therefore, individual variances are inevitable in these cases. We are confident that for all 15 participants, all tone-discriminant electrodes and the majority of the speech responsive electrodes are located either within STG or close to the boundary of STG.

Edits: new Fig S3 after correcting the localization error (see below).

(3) Abstract and main text: Though the abstract infers that analysis was systematically performed in two distinct patient groups (i.e., Mandarin/intraoperative recording vs. English/bedside recording), we found that some analysis was performed only in one of the two groups (or the results were described for only one group). Thus, we are not sure whether the results of this study fully support all of the conclusions.

The main theme of the first part of the manuscript is about the encoding of lexical tones in native mandarin speakers, therefore we did not focus on native English speakers. However, we replicated the main findings (i.e. Fig 3) in the English group as well, but left it as supplementary results (see Fig S7). We edited the text in the Results section to emphasize that we found consistent encoding properties in both Mandarin and English groups.

The major part of cross-language comparisons are the results presented in Figs 4&5, which were performed in both groups. We also included a replication of the original Figure S4 (now Fig S10) on English corpus to address the concern raised by the reviewer. In particular, we found that both the encoding model trained on Mandarin speech and the encoding model trained on English speech can predict the neural

response to English speech. We also included this analysis into the supplement information (Fig S11).

On a side note, pitch representation in non-tonal language (e.g. English) has been studied in previous literature (Tang et al. 2017) and is not the main focus of this paper. Pitch mainly represents intonational information in English speech and there is no English counterpart of lexical tones, so we did not study the categorical representation of pitch in English speech.

Edits:

1. We have now included new supplement figures: Figs S8 and S11.
2. Page 6, Line 188-191: "Moreover, we found consistent encoding properties when similar analysis was applied to STG electrodes in non-tonal language speakers when they listen to Mandarin speech (Fig. S7), suggesting a general auditory mechanism for pitch processing."

(4) Methods/Discussion: It is unclear how they controlled the effects of different experimental conditions on high-gamma activity. Mandarin speakers were given auditory stimuli during the awake craniotomy, whereas English speakers underwent the task at the bedside. Mandarin speakers had brain tumors, whereas English speakers had drug-resistant epilepsy. Some Mandarin speakers had electrodes placed on the right hemisphere, whereas English speakers had electrodes exclusively on the left hemisphere.

We agree that this is an important concern when performing cross-group comparisons. We have tried our best to rule out potential confounding due to the differences in population groups. In particular, 1) the two groups of patients used the same type of high-density grids, amplifier and digital signal recording system; 2) all patients have normal hearing and language ability in their native languages; 3) we only include patients whose tumor or epileptic foci was not in the lateral temporal cortex; 4) the same results were consistently found when we constrained the analysis only to the left hemisphere or to the right hemisphere (Fig S7-9).

(5) Methods/Discussion: It is unclear whether the study included bilingual patients. This information may be useful to determine whether the high-gamma patterns are indeed tuned to a given language.

No participant is considered as bilingual. We have added this note into the main text.

Edits: Page 3, Lines 68-69: "... (11 native Mandarin speakers, and 4 native English speakers; all are monolingual speakers)..."

(6) Methods: The authors should provide the specifications of high-density electrodes used for each patient group.

High-density electrode grids (Integra or PMT) with identical specifications (4mm center-to-center spacing and 1.17 mm diameter exposed contact lateral) were used in both groups. Depending on the exact clinical need, the grid may have 128 (16 x 8) or 256 (16 x 16) contact channels in total.

Edits: we have added the above information to the Methods section (Page 17, Line 516-519).

(7) Methods/Supplementary document: The authors should describe the specifications (e.g., number of syllables) of Mandarin and English auditory stimuli.

The ASCCD set consists of 68 passages of Mandarin speech selected from the ASCCD corpus, spoken by 10 different speakers (5 males, 5 females). The length of single passage varies between 10 to 60 sec. Across all passages, we have a total of 4711 tokens (syllables) of the four lexical tones. The passages were separated with 0.5 sec of silence.

The BURSC set consists of 62 passages of English speech selected from the BURSC corpus, spoken by 6 different speakers (3 males, 3 females). The length of single passage varies between 10 to 60 sec. The passages were separated with 0.7 sec of silence.

The TIMIT set consists of 499 English sentences selected from the TIMIT corpus, spoken by 402 different speakers (286 males and 116 females). The sentences were separated with 0.4 sec of silence.

Edits: the above information is added into the Methods section on Page 18.

(8) Methods: The authors should define “specific recording block” on Line 510.

Edits:

Page 17, Line 532 “... The tasks were broken into recording blocks of ~5 minutes in length. The high-gamma signal was z-scored across the recording block.”

(9) Methods: The authors should explicitly describe the electrode montage/reference used for intracranial recording.

Edits:

Page 17 Line 527: “(the signal was) ... re-referenced to the common average across channels sharing the same connector to the preamplifier. “

(10) Methods: The authors should provide the number of speech responsive STG electrodes in ‘pitch temporal receptive field analysis’ and ‘population tone decoding and representational similarity analysis.’

We have added the corresponding detailed information to the main text and Methods section.

Edits:

1. Page 6 Line 167: “Out of 541 speech responsive electrodes across all 11 native Mandarin-speaking participants...”
2. Page 6 Line 189: “...we found consistent encoding properties when similar analysis was applied to 183 speech-responsive STG electrodes in English speakers when they listen to Mandarin speech (Fig. S7)... ”
3. Page 21 Line 673: “All speech-responsive electrodes in Mandarin (M1-M11) and English (E1-E4) speakers were included in the TRF analysis.”
4. Page 22 Line 763: “To maximize the statistical power, we only included speech-responsive electrodes in subjects that finished all ASCCD blocks (M1-M7 and E1-E3).”
5. Page 13 Line 358-359: “Specifically, we pooled speech-responsive electrodes in the non-primary auditory cortex across subjects in this analysis (316 in Mandarin participants and 171 in English participants).”

Reviewer #2 (Remarks to the Author):

This study explores the auditory and linguistic representation of Mandarin lexical tones in superior temporal gyrus using ECoG. The data show strong and convincing evidence for *auditory* encoding of lexical tones based on speaker-normalized relative pitch height and pitch change. Evidence for *linguistic* processing of lexical tones in STG is sought (and found) in the comparison of Mandarin vs. English speakers' neural responses to Mandarin speech. Results converge across a number of cutting-edge analyses of high gamma activity, including modelling of single electrode data with temporal response functions (TRFs) within Mandarin and across Mandarin and English speech, as well as multivariate pattern classification and representational similarity analyses at population level.

To my knowledge, this is to date the most compelling study on the encoding of lexical tone, and it integrates well into previous work of the same lab on categorical phoneme perception (Chang et al., 2010, Nat Neurosci) and intonation perception (Tang et al., 2017, Science) in non-tonal languages. Results are novel, will influence thinking in the field, and are also of interest to a broader readership, given that about half of the world's languages are tonal. Authors succeed in providing within-study replication of results by complementary analyses, all results are corrected for multiple comparisons.

Still, I do have a few major points that the authors may wish to address in a revision to allow other researchers to reproduce their work and to better link the results to prevailing language models. More precisely, the choice of acoustic features (e.g., the exclusion of spectrum), their extraction from the speech signal, and the construction of TRF models need better motivation /

specification (major points [1] to [3]). Moreover, given that Mandarin (but not English speakers) comprehend Mandarin speech, it should be discussed whether STG truly encodes linguistic tone categories, or whether results reflect the top-down modulation of auditory processes by remote linguistic processes (major points [4] and [5]).

We thank the reviewer for the supportive comments on the significance of the manuscript. We have responded to the comments point-to-point below. We feel that addressing these comments have improved the quality of the manuscript. We are particularly grateful for the detailed comments that help improve the clarity of the work.

Major points:

[1] Extraction of acoustic features (l. 83-100). The methods leading to conclude that “relative pitch height and pitch change are speaker-normalized acoustic features that define lexical tones in Mandarin” (l. 83-84) are hardly presented in the Methods section. Please add a description to the Methods of how relative pitch height and pitch change were calculated (as shown in Figure 1). Also the PCA, its execution and results need more explanation. What did you enter into the analysis? Did you calculate one PCA for each time point? Does “average” in S1C and S1D mean “average across exemplars from each tone”? Did you use Praat for pitch extraction? How many tone exemplars were entered into the analyses, etc.?

We have added the corresponding information to the Methods section and to the caption of Fig S1. We have also included new Fig S4 to demonstrate the extracted features.

Edits:

1. Page 18 Lines 570-593:

Spectrum and pitch feature extraction

The spectrum features of the speech were calculated using a mel-band auditory filterbank of 30 filters ranging from 0 to 8KHz. Absolute pitch was calculated using procedures identical to Tang et al.18, where the fundamental frequency (F0) was calculated using an automated autocorrelation method in Praat and corrected for halving and doubling errors. The absolute pitch was defined as the natural logarithm of F0 values in Hz. The relative pitch was computed by z-scoring the absolute pitch values (log F0) within each sentence/passage (within speaker). The pitch change was computed by taking the first-order derivative (finite difference) in time for logF0. We discretized absolute pitch, relative pitch and pitch change into 10 bins, equally spaced from the 2.5 percentile to the 97.5 percentile value. The bottom and top 2.5% of the values were placed into the bottom and top bins respectively. As a result, absolute pitch, relative pitch and pitch change were represented as three 10-dimensional binary feature vectors.

Principal component analysis of lexical tones in acoustic pitch space

We used principal component analysis (PCA) to analyze the acoustic pitch space of lexical tones. We extracted the pitch contours of the 4611 tone exemplars in the ASCCD corpus. These contours were time-warped to have the same length of 250 ms (25 time points at 100 Hz sampling rate). PCA was performed on this 4611 x 25 data matrix X , and we got decomposition $X = LWT$, where L is a 4611 x 25 PC score matrix, and W is a 25 x 25 orthogonal weight matrix with columns of W form an orthogonal basis for the 25 temporal features.

2. Supplement Figure S1(C) Each dot is a single tone exemplar, x-axis corresponds to the PC score (loading) along the first PC dimension, and y-axis corresponds to the average relative pitch height of the exemplar across time.
3. Supplement Figure S1(D) Each dot is a single tone exemplar, x-axis corresponds to the PC score (loading) along the second PC dimension, and y-axis corresponds to the average pitch change of the exemplar across time.
4. Supplement Figure S4 is added to demonstrate the extracted features.

[2] Choice of acoustic features – exclusion of spectrum (l. 102-187). The Methods section should specify how / with which software spectrum and intensity were extracted and how TRF models were built. What does “spectrum” refer to – the spectral centroid, SD of spectrum or else? Judging from the unique R2 values (lines 147-148), spectrum actually looks like a very good predictor – even better than speaker-normalized features. How would spectrum compare to speaker-normalized features in Fig. 3, and what does it mean for the conclusion that “the differential neural responses with regard to lexical tones are ‘mainly’ driven by the encoding of speaker-normalized pitch features” (lines 183-184)?

The spectrum features were calculated using a mel-band auditory filterbank of 30 filters ranging from 0 to 8KHz.

To address the reviewer’s concern, here we did a comparison between feature encoding of spectrum and pitch. Although a lot of the electrodes were showing significant unique R2 for spectrogram features, the encoding of spectrogram features and the speaker-normalized pitch features were not correlated in the pitch encoding populations (Pearson’s $r = 0.11$, $p = 0.2$). Furthermore, the encoding of spectrogram features did not show significant correlation to the tone discrimination in individual electrodes (see right panel of Fig S6). Compared to Fig. 3D, we draw the conclusion that the differential neural responses with regard to lexical tones (the components that are discriminant between tones) are mainly driven by the encoding of speaker-normalized pitch features.

One of the main reasons to include the spectrum features is to use it as a control for all the other factors in the speech that are not included in the feature sets. It has been shown in previous ECoG studies that the speech responsive electrodes in STG are

sensitive to spectrogram feature patterns (e.g. Mesgarani et al. Science 2014). Therefore, it is not surprising that it contributes significant unique R2. However, as shown in Fig 2 and 3D, the spectrum (+intensity and absolute pitch) does account for the overall mean speech responsive activity, but it is the speaker-normalized pitch features that actually account for the tone discriminant patterns in neural response.

Edits:

1. Page 18 Line 572: “The spectrum features of the speech were calculated using a mel-band auditory filterbank of 30 filters ranging from 0 to 8KHz.”
2. Page 6 Line 183-186: “Therefore, at the level of individual electrodes, the differential neural responses with regard to lexical tones are mainly driven by the encoding of speaker-normalized pitch features rather than absolute pitch or other lower-level acoustic features (Fig. S6).”

Figure S6. Encoding of speaker-normalized relative pitch height and pitch change versus spectrogram features. Left: Scatterplot of the unique variance explained by spectrogram features and by speaker-normalized pitch features, across speech-responsive electrodes from all Mandarin-speaking participants (M1-M11). Each dot represents a single electrode. Colored dots indicate significant encoding of speaker-normalized pitch (red; $p < 0.01$, permutation test). No significant correlation was found between spectrogram encoding and speaker-normalized pitch encoding. Right: Unique variance explained by speaker-normalized pitch features and tone discriminability. Red dots indicate electrodes that had significant encoding of speaker-normalized pitch features ($p < 0.01$, permutation test). No significant correlation was found between spectrogram encoding and tone discrimination.

[3] TRF models (l. 139-187). How exactly were TRF models built, e.g., how were non-pitch periods (consonants, pauses etc.) coded? A supplementary figure would help (e.g., like Fig. 1A in Di Liberto et al., 2020, eLIFE), along with a more detailed description in the Methods section.

We have now included detailed information about how the features are extracted and how the TRF model is defined and fitted. We also included a supplement figure for visualization of the extracted features.

Edits:

1. Page 18 Line 572-584:

Spectrum and pitch feature extraction

The spectrum features of the speech were calculated using a mel-band auditory filterbank of 30 filters ranging from 0 to 8KHz. Absolute pitch was calculated using procedures identical to Tang et al., where the fundamental frequency (F0) was calculated using an automated autocorrelation method in Praat and corrected for halving and doubling errors. The absolute pitch was defined as the natural logarithm of F0 values in Hz. The relative pitch was computed by z-scoring the absolute pitch values (log F0) within each sentence/passage (within speaker). The pitch change was computed by taking the first-order derivative (finite difference) in time for logF0. We discretized absolute pitch, relative pitch and pitch change into 10 bins, equally spaced from the 2.5 percentile to the 97.5 percentile value. The bottom and top 2.5% of the values were placed into the bottom and top bins respectively. As a result, absolute pitch, relative pitch and pitch change were represented as three 10-dimensional binary feature vectors. For non-pitch periods, these feature vectors would have all 0s. See Fig S4 for a demonstration of feature extraction.

2. Page 20 Lines 635-639:

Temporal receptive field (TRF) models allow us to predict neural activity based on stimulus features in a window of time preceding neural activity. In particular, we fit the linear model $y(t) = \sum_{f=1}^F \sum_{\tau=0}^T \beta_f^T(\tau) x_f(t - \tau) + \epsilon$ for each electrode, where y is the high-gamma activity recorded from the electrode, $x_f(t - \tau)$ is the stimulus representation vector of feature set f at time $t - \tau$, $\beta_f(\tau)$ is the regression weights for feature set f at time lag τ , and ϵ is the gaussian noise.

3. Supplementary Figure S4:

Figure S4. Examples of feature extraction. Left and right columns represent the same sentence spoken by two different speakers (left column is a female speaker, right column is male) and different set of features all aligned to the same time. First row: raw waveform; second row: high-gamma (z-scored) activity at an example electrode; third row: Mel-scaled spectrogram; fourth row: absolute pitch (binned into 10 bins); fifth row: relative pitch (binned into 10 bins); last row: pitch change (binned into 10 bins).

[4] Interpretation of neural tuning shaped by language experience (l. 300-338). I find it surprising to see that integration windows were *shorter* in English than Mandarin listeners (Figure 5C). Given that pitch is a suprasegmental feature in non-tonal languages, typically spanning several syllables, wouldn't one expect to see *longer* integration windows in English than Mandarin listeners? Hence, are these integration differences driven by "statistical differences" between English and Mandarin, or by the fact that Mandarin (but not English) speakers comprehend the presented material, triggering top-down linguistic processes? Please add a comment on that in the manuscript. [One way to experimentally address this question may be to investigate whether integration windows in English listeners are longer when listening to their native English language than to Mandarin speech. Likewise, to dissociate the influence of speech statistics and top-down comprehension related processes, it may be interesting to add another language with similar statistics, but that listeners don't understand (e.g., German or Thai or tonal chimeras/jabberwocky) in future work.]

It is not likely that this is purely an effect of understanding the materials, since cross-linguistic models trained on English and Mandarin corpora showed consistent patterns (Figs 4, S9-11; also see responses to R1 #3). As the reviewer suggested, we also computed the mean integration window on English corpus for Mandarin speakers and English speakers to validate this (see the attached figure below). Because we have less blocks of English data, the patterns look slightly different from the TRF models fitted on the Mandarin corpus (Fig 5), but we still found that Mandarin speakers (red curves) showed longer integration (250-280 ms) than English speakers (blue curves) in relative pitch height, and no significant difference in pitch change.

Given that we don't see a pattern change in the mean TRF, especially the pattern in the late time period (e.g. after 200ms when the top-down effect is likely to take place) across languages, it is not likely that it is purely a top-down effect.

That being said, we are only recording from STG, therefore we cannot fully exclude the alternative of "top-down" linguistic processes with influence from higher-order areas (e.g. frontal lobe). However, the bottom line is that our results indicate that STG can be involved in these linguistically-related processes, although whether or not it emerges from computations within STG or through interactions in the broader language network remains unclear.

Edits:

Page 15 Line 463-468 "However, it is worth pointing out that due to the limit in the coverage of ECoG, we cannot evaluate the long-range fronto-temporo-parietal connections or the potential involvement of other areas during the linguistic perception processes^{13,41,46}. Therefore, it remains unclear whether the linguistically relevant categorical representation emerges from recurrent interactions within the non-primary auditory cortex or is a result of reciprocal top-down influence from other interacting areas in the language network."

[5] Discussion. “Therefore, our results provide evidence that STG is involved in both general auditory and *linguistic processing*.” (l. 451-452); “Speech perception relies upon a shared auditory feature processing mechanism in the cortex, which is tuned to the *statistics of a given language*.” (abstract). These conclusions / wordings seem too strong. The fact that neural activity discriminated tone categories to a significant degree also in English native speakers indicates that processes are primarily auditory rather than linguistic (Figure S5, purple bars in Figure 5H), as acknowledged by the authors. An explanation that should be considered is the influence of top-down linguistic processes due to language experience/comprehension that drove/shaped auditory processes in STG via long-range fronto-temporo-parietal connections in Mandarin speakers (see also comment [3]).

We agree with the reviewer that our data cannot fully exclude this alternative explanation. We have added the limit and alternative explanations to the discussion. See the above point.

Minor points:

- General: It would be helpful if the authors indicated in the Methods which analyses correspond to which Figure(s) or Figure panels.

We have added extra information indicating with Figures the analyses were corresponding to.

- General: Methods switch between past and present tense; tense could be more consistent.

We thank the reviewer for pointing this out. We have made the tense consistent.

- L. 479-494: Please add a note on whether Mandarin (English) participants were able to comprehend the English (Mandarin) material or not.

All participants are considered monolingual. See also R1 #5.

Edits: Page 3, Line 68-69: “...(11 native Mandarin speakers, and 4 native English speakers; all are monolingual speakers)...”

- L. 556: typo – change “discriminant” to “discriminate”.

Corrected.

- L. 554-562: Please specify whether all electrodes or only speech responsive electrodes were inspected in this analysis?

Only speech responsive electrodes were inspected.

Edits: Page 19 Line 622: “To find speech responsive electrodes that also discriminate between lexical tones...”

- L. 118: typo – change “Fig. 2(B, H)” to “Fig. 2(B, G)”.

Corrected.

- L. 587-592: “We shuffled the acoustic features between all phrases ...” – what is meant by phrases?

Edits:

Page 20 Lines 658-662: “When performing the permutation test, we would like to keep the intrinsic temporal structure in the speech stimuli while permuting the correspondence between the stimuli and neural activity. Therefore, we found phrases in the continuous speech and shuffled at the level of phrases. Here a phrase is defined as a continuous speech segment with at least 150 ms of silence before and after. “

- L. 164-175: Please specify in the Methods section that t-tests were run only on unique R2 values of electrodes that were responsive either to speaker-normalized pitch OR absolute pitch (tone category). Those that responded to both were excluded, weren't they?

We apologize for not making it clear. We actually ran the t-test on electrodes that were responsive to either speaker-normalized pitch or absolute pitch (tone category), including those that responded to both.

Edits:

P20 Line 668-671: “To compare the representation of speaker-normalized pitch and absolute pitch (or tone category), we used paired t-test to compare the unique R2 for speaker-normalized pitch against the unique R2 absolute pitch (or tone category) on electrodes that were significant for either speaker-normalized pitch or absolute pitch (tone category), including those that were significant to both.”

- L. 180-181: Please specify what “F-statistics across time” means. Mean or median F-value across time, and in which time-window relative to tone onset?

Edits: we have changed it into “the maximum F-statistics across the 1s time window after tone onsets”

- L. 234: typo – change “Figure 4A” to “Figure 4B”.

Corrected.

- L. 240: typo – change “(Fig. S3G, H)” to “(Fig. S4G)”.

Corrected.

- L. 248-252: “To quantify this similarity across all neural populations ...” and “For nearly all electrodes ...” – does this mean all electrodes, all speech-responsive or all tone-responsive electrodes?

Thanks for pointing out the ambiguity. We are referring to all tone-responsive electrodes.

Edits: Page 9, Line 251-254: “To quantify this similarity across all tone-discriminating neural populations, we calculated the correlation between the regression weights for the Mandarin and English models within the same tone-responsive electrode (Fig. 4G).”

- L. 616: “we first spatially blurred the weights” – how was this done?

Edits:

Page 21 Line 698-700: “To reduce noise in the weights, we first spatially blurred the 2D weight matrices (# of time lags by # of feature dimensions) using a 2D-gaussian filter with sigma of 1 (the filter was applied to each feature group separately), and then calculated the correlation.”

- L. 272-275: “Mandarin/English group” and “within Mandarin group” sounds like referring to participant groups (Mandarin and English native speakers). Please rephrase.

We have changed them into “across languages same electrode” comparison and “same language across electrodes” comparison

Edits:

Page 10 Line 275-279: “The “across languages same electrode” comparison (right) shows significant positive correlations between the weights from the Mandarin model and English model within the same tone discriminating electrode. The baseline distribution of the “same language across electrodes” comparison (left) shows no significant correlation between the regression weights from the Mandarin model of different tone discriminating electrodes”

- Figure S4 G-I: Given that “Mandarin tone contours have a larger dynamic range of pitch and are temporally organized by syllable-sized segments compared to English speech” (lines 302-303), it seems rather natural/expected that the Mandarin model outperforms the English model. What is it that the authors intend to say with these comparisons?

The reviewer is correct that these are expected. What we are actually trying to convey is that although there is a small performance gap, the cross-language prediction (use English model to predict Mandarin response) is actually quite reliable. We have removed the statement about the difference between model predictions in Fig S10 and

emphasizing the similarity between the two by adding an additional Fig. S11. See also R1 #3.

- L. 623: Does “speech-responsive STG electrodes” mean that speech-responsive electrodes outside of STG were excluded?

To maximize the statistical power, this analysis only included subjects who finished all 6 blocks of the Mandarin task (M1-M7 and E1-E3). In these subjects, all speech-responsive electrodes were either located within STG or near the boundary of STG (Figure S3). (see also R1 #2)

- L. 636 and 640: “in the STG network” vs. “all speech responsive electrodes” – please specify because not all speech responsive electrodes were located in STG.

Similar to the previous point, here we replace “STG network” with “speech-responsive STG network”. In fact, as we mentioned in response to R1 #2, we corrected an error in the previous version of the manuscript that caused some speech-responsive electrodes to be located outside of STG (in M9 and M11; see the corrected Fig S3; see also R1 #2).

- L. 640: “neural activity across all speech responsive electrodes” – please specify that/whether “neural activity” refers to mean HG in the 50ms window, per electrode.

We are concatenating all 5 time points within the 50 ms window as features.

Edits: Page 22 Lines 727-728: “the neural activity across the 5 consecutive time points in all speech responsive electrodes were concatenated and used as features to train a pattern classifier.”

- L. 352: What does “peak” in “peak overall pairwise classification accuracy” refer to? The sliding window, i.e. time? Or tone pair? Both?

We added the following explanation to the Methods.

Edits:

Page 22 Line 731-733: “We first computed the average pairwise classification accuracy across all 6 pairs for each sliding time window, and then took the maximum accuracy across all sliding windows as the “peak overall pairwise classification accuracy”.”

- L. 660 and legend of Figure S5: What is “STG space”? More generally, the second half of the manuscript talks about STG activity patterns, STG space and similar, i.e., an anatomical choice of electrodes in contrast to the choice based on functional speech-responsiveness at the beginning of the manuscript.

Similar to the above points, as well as to R1 #2, all of the speech responsive electrodes were located within or near the boundary of STG. Therefore we used “speech-responsive” and “STG” interchangeably. To avoid confusion, we have changed the terminology into “speech-responsive STG population” to reflect both the anatomical (superior temporal gyrus) and functional (speech-responsiveness) of the neural population that we are interested in.

- Figure S9 needs more explanation. What does it show? How was it obtained? How does it show that tone 3 is less discriminable from the other tones in English than Mandarin native speakers?

We have added detailed descriptions to the caption of Fig S14 (originally Fig S9).

- L. 466: “which is extracted in a context-dependent way” – please specify what “context” means.

By “context-dependent” we were referring to the fact that speaker-normalization would require the listener to be aware of the “context” of the absolute pitch.

Reviewer #3 (Remarks to the Author):

In this manuscript from Li et al, the authors investigate how lexical tones present in Mandarin speech are represented in human non-primary auditory cortex, using high-density ECoG recordings. They find that high gamma-activity in individual electrodes can encode features relevant for discriminating between lexical tones (speaker normalized pitch height and change). Using models based on English rather than Mandarin produced similar regression weights (positive correlation) across electrodes, suggesting speaker-normalized pitch encoding was a general feature of non-primary auditory cortex, and did not require a tonal language. Furthermore, native English speakers also could represent pitch height and change, like Mandarin speakers. But while pitch change showed similar neural tuning, pitch height tuning had a greater dynamic range in native Mandarin speakers (compared to native English speakers), likely due to the different use of pitch in these two languages.

The experiments are well designed and thoughtfully analysed, and the results are an important step forward for the field. I only have a few minor comments aimed to improve the clarity and overall understanding of the results.

We thank the reviewer for the supportive comments. We have responded to the comments point-to-point below.

Minor comments:

1. In fig 2B/G- actual HG zscore never goes above 1, even though the activity is significant different between tones. I assume this is due to the z-score being calculated over the entire recording block. If instead you calculate the z-score using the mean and standard deviation of the recording period when no sound was present (which presumably would have less neural activity), would the zscore be significantly different from baseline? In other words, are the neurons driven by the stimulus?

There are two main reasons for this: (1) The reviewer is correct that we z-scored over the entire recording block, hence resulted in a larger estimation of the variance than a resting-state baseline; (2) we should point out that Fig 2B/G are averaged ERPs of single tones, which are computed by averaging across all tone tokens in the entire speech. Therefore, the majority of these tokens are located in the middle of sentences, where there is a general decay in the amplitude of neural response, compared to the onsets of the sentences (e.g. see Hamilton et al., Current Biology 2018).

To better demonstrate the speech responsiveness of these populations, we show the average ERP aligned to sentence onsets and z-scored to pre-onset baseline period (with significant electrodes marked as red). Here the maximum z-score reaches 5. (See the figure below)

2. Fig 2E (complete model) looks much more like the real data, but is due to greater complexity of the model. If you remove parameters such as spectrum, intensity and absolute pitch, does it still match? Can you resolve whether a better fit is due to just greater complexity or using the correct parameters.

In Fig 2, the model is fit with all features included together at once, and the prediction is plotted using the weights of a subset of features at each step, so we are seeing something roughly equivalent to “unique contribution” in prediction. Therefore, model complexity is not a concern for this case.

3. Line 121- “This shows that adjacent STG electrodes may be differentially tuned to lexical tones.” Are you suggesting that there is a topographical organization for lexical tone in the STG? Or is there another explanation for this result? If the neurons that underlie the recorded gamma signal are heterogenous in how they respond to lexical tones (e.g. half the neurons prefer T4>T1 and the other half prefer T1>T4), is this detectable with current methods.

Our results suggest that at the level of high-density electrodes in STG, we did not find populations selective to a single lexical tone, but they are rather tuned to the speaker-normalized pitch features that underlie the tone discriminant patterns. However, it is also possible that at a finer scale (e.g. single neuron/neuronal column level) the neurons may behave differently, as the reviewer suggested. This is a limit of the current technique that we can only look at the level of single electrodes. Future studies with finer grain recordings may be able to resolve the question.

4. Generally, some more raw data would help. It’s good that you show the actual HG plots in Fig 2B and G, and where these electrodes were located (in A) but it would be helpful to see how this looks across all the electrodes (and highlight the specific time-amplitude plots that are significant and non-significant).

We have added a supplement figure (Fig. S2) showing the raw ERP for the entire grid in Fig 2.

5. Line 168- Speaker normalized pitch features explained up to 6% variance. Is this a lot? You write on line 183- “Therefore, at the level of individual electrodes, the differential neural responses with regard to lexical tones are mainly driven by the encoding of speaker-normalized pitch features.” This seems to contradict the 6%...

The 6% is the unique variance explained only by speaker-normalized pitch features. The features (spectrogram, intensity, absolute pitch, speaker-normalized pitch) were not entirely independent from each other, therefore the unique variance is not the actual amount of total variance that correlates to the feature. If we only include speaker-normalized pitch features in the encoding model, the R2 can go up to ~0.15 – 0.2 (i.e. 15-20% of the total variance). It is true that other features, such as spectrogram features, also explained a significant amount of unique variance in the model. We are not trying to argue that these electrodes only responded to speaker-normalized pitch and not to other speech features. Indeed, as shown in Fig 2 and 3D, the spectrum (+intensity and absolute pitch) does account for the overall mean speech responsive activity, but it is the speaker-normalized pitch features that actually explain the tone discriminant patterns in neural response. (See also response to R2 #2).

referee- Daniel Bendor

Reviewer #4 (Remarks to the Author):

The authors study the pitch encoding properties of the superior temporal gyrus (STG), as a key non-primary auditory cortical area, across native Mandarin and English speakers using electrocorticography (ECoG). They consider two main speaker-normalized pitch features, namely relative pitch height and pitch change, and demonstrate the differential encoding of these features in STG across the two speaker populations.

The first finding of the study is that in Mandarin speakers, tone category, spectrum, intensity and absolute pitch are not differentially encoded across the 4 tones in Mandarin, whereas relative pitch height and pitch change are. The second finding is that the encoding of relative pitch height and pitch change actually contribute to the discriminability of the tone categories (which is not the case for absolute pitch). Third, it is shown that the encoding models of these speaker-normalized pitch features are remarkably similar across English and Mandarin speech, suggesting a universal neural representation of these features. Fourth, using temporal receptive field (TRF) analysis, the temporal profile of the neural responses of native Mandarin and English speakers to Mandarin speech are shown to pertain to different time-scales, perhaps tuned to the language-specific statistics of tone contours. Finally, it is shown that while these speaker-normalized features are encoded at the electrode level (but not tone categories), there is a population-level emergence of categorical representation of the tones in STG.

The paper is well-written overall, the methods are compelling, and the results are significant, as they provide new evidence on the role of STG in concurrent processing of both universal and language-specific pitch features. I have the following comments that would like the authors to address:

We thank the reviewer for the supportive comments recognizing the significance of our results. We have responded to the comments point-to-point below.

Major comments:

1) The emergence of the population encoding of tone categories is an impressive finding. It would be helpful to visualize the weights of the multi-variate classifier (logistic regression in this case), and see if the spatial/spatiotemporal nature of these coefficients can provide further anatomical insights on the underlying population neural code across STG. For instance, is the population code the result of positive posterior and negative anterior weights in the STG electrodes? I understand that the pitch-selective electrodes are pooled across subjects to increase statistical power, and this may hinder finding a clear anatomical structure in the regression coefficients, but probing the anatomical structure of the classifier weights would be helpful to shed light into the nature of the population code.

By plotting the average between tone classifier weights for each individual electrode on a warped common brain, we found no apparent patterns in the weight distribution. Most of the electrodes with larger weights localized in mid/post STG as we expected. The positive and negative weights were also distributed across the entire STG.

We should note that due to variations in the intro-operative localization and warping, the exact location of these electrodes may not be accurate in this common brain (see Figure S3 for localization in individual spaces). The other thing we want to point out is that in individual cortices in these subjects, the tone-discriminant electrodes also tend to cluster in mid/post STG (Fig S3).

Edits: this is included as supplementary Figure S15.

Figure S15. Visualizing the averaged between tone classification weights in a warped common space for Mandarin speakers. The averaged between-tone classification weights (averaged across all 6 pair-wise classifiers) at peak time of classification accuracy were plotted on a common brain for all speech-responsive electrodes. Blue color indicates positive weights and red color indicates negative weights. Darker color indicates smaller absolute value and black corresponds to 0.

2) On a related note, lines 641-642 mention that the pitch-encoding electrodes were pooled across subjects to construct the feature space, which is totally fine. But, in reading pages 13-14 (regarding Fig. 5F and 5G), I was under the impression that the categorical indices were computed at the individual subject level and then averaged to get panels F and H. I suggest that the authors make this explicit on page 13 that the results follow pooling all the pitch-encoding electrodes across subjects.

We thank the reviewer for pointing this out. We have now made it explicit in the Results section as well as in the Methods.

Edits:

1. Page 13 Line 358-361 “Specifically, we pooled speech-responsive electrodes in the non-primary auditory cortex across subjects in this analysis (316 in Mandarin participants and 171 in English participants). We computed the peak overall pairwise classification accuracy between the concatenated population neural responses to the four lexical tones using a multivariate pattern classifier...”

2. Page 13 Line 386-387 "...evaluate the representational similarity between different groups of syllables in the concatenated neural population space and acoustic space, respectively."

3) Fig. 5, Panel H suggests that the categorical index based on the pitch change partition is higher than that corresponding to relative pitch height partition for Mandarin speakers (which is not the case for English speakers). This suggests that relative pitch height has a smaller role in the population-level categorical encoding of tones compared to the pitch change for Mandarin speakers. But, at the same time panel E shows very similar TRFs for the pitch change across English and Mandarin speakers, whereas the TRFs for the relative pitch height in panel C are quite distinct. Judging by panels C and E, one would expect that the categorical index for the relative pitch height partition would also be higher than that of pitch change (which is the opposite of what panel H shows). Do the authors have any thoughts on/explanations for this seeming discrepancy?

Partition is not directly related to tuning and categorical sensitivity. It is indeed the opposite of what the reviewer was suggesting. To see this, the partitions are done within each tone category, therefore, for the pitch height partition, each tone will have high pitch subgroup and low pitch subgroup, this attenuated the between tone difference in pitch height (e.g. the low pitch subgroup in Tone 1 is more similar to the subgroups in Tone 3 in terms of pitch height) and would require the classifier to seek for additional features (e.g. pitch change coding) to resolve the classification. Similarly, pitch change partition reduces the between group differences in pitch change and requires the classifier to use additional features (e.g. pitch height) to resolve the decoding.

4) The description of group-lasso on Page 19, lines 647-650 is not accurate. I believe the authors are referring to grouping the coefficients of each electrode in the logistic regression by the L2-norm and summing the L2-norms to induce an L1-norm penalty across electrodes. If this is the case, I suggest the following rewording: "...where L2-norms of all the temporal coefficients from each electrode were summed to induce an L1-norm penalty across electrodes". Also, the statement "network interactions between local populations" is not accurate. I suggest the following rewording: "...while maintaining temporal smoothness within electrode and promoting sparse interactions between electrodes".

We thank the reviewer for the suggestion. We have made the suggested changes to the Methods section.

5) I noticed that several references to figure panels of the main text and supplementary material were mis-numbered:

- Page 4, Line 118: Fig. 2(B,H) -> Fig. 2(B,G)

- Page 9, line 234: Fig. 4A -> Fig. 4B

- Page 9, line 240: Fig. S3G,H -> Fig. S4G,H,I.

- Page 13, line 359: Fig. S6 -> Fig. S5

We thank the reviewer for pointing these out. We have made the edits in the main text.

Minor comments:

6) Page 4, lines 119-121: the statement that the first electrode responded significantly to tone 1, 2 and 3 is not accurate. It seems that the electrode responds to all tones significantly, if significance is assessed based on the pre-onset response. Perhaps you can just say the first electrode shows higher responses to tones 1, 2, and 3.

We have made the suggested changes.

7) Some typographical/wording suggestions:

- Abstract: auditory speech cortex -> auditory cortex

- Page 4, lines 93-94: speaker-normalized pitch height -> speaker-normalized relative pitch height

- Page 9, line 234: English-data -> English data

- Page 12, line 331: $p < 0.05$ -> $\alpha < 0.05$ (I assume 0.05 is the target FDR level, as in the caption of Fig. 5C).

We have made the suggested changes.

REVIEWERS' COMMENTS

Reviewer #1 (Remarks to the Author):

The authors satisfactorily addressed all issues raised by this reviewer. We want to congratulate their effort and hope they will continue enjoying the research activities.

Reviewer #2 (Remarks to the Author):

Thank you to the authors for considering all of my concerns so carefully. All points have been sufficiently clarified, and I have no further major comments.

Minor points:

- Line 229: The opening of this paragraph needs a bit of an adjustment after having inserted the new sentence in lines 189-191.
- Fig. S14: For consistency with Figure 5G, numbers could range from 1-4, instead of 0-3.
- Lines 591: Should it be "4711" or "4611"?
- Please have the manuscript checked again for consistency in tense (e.g., lines 106, 190, 233, 377, 542-546, 549-555, 734, 748, etc.).

signed: Daniela Sammler

Reviewer #3 (Remarks to the Author):

I'm happy with all the revisions and support publication. Great job.

Reviewer #4 (Remarks to the Author):

The authors have done a great job clarifying and addressing my previous questions and comments. I have no further comments.

We thank all reviewers for their insightful and supportive comments during the review process. We have made the suggested edits by R2. The new edits are highlighted in blue color in the manuscript.

REVIEWERS' COMMENTS

Reviewer #1 (Remarks to the Author):

The authors satisfactorily addressed all issues raised by this reviewer. We want to congratulate their effort and hope they will continue enjoying the research activities.

Reviewer #2 (Remarks to the Author):

Thank you to the authors for considering all of my concerns so carefully. All points have been sufficiently clarified, and I have no further major comments.

Minor points:

- Line 229: The opening of this paragraph needs a bit of an adjustment after having inserted the new sentence in lines 189-191.

We have edited the paragraph opening (now in lines 179-181) and the new inserted sentence (now in lines 172-176).

- Fig. S14: For consistency with Figure 5G, numbers could range from 1-4, instead of 0-3.

We have made the suggested edits.

- Lines 591: Should it be “4711” or “4611”?

We have changed it to “4711”.

- Please have the manuscript checked again for consistency in tense (e.g., lines 106, 190, 233, 377, 542-546, 549-555, 734, 748, etc.).

We have corrected the tense in these sentences and in other places in the manuscript.

signed: Daniela Sammler

Reviewer #3 (Remarks to the Author):

I'm happy with all the revisions and support publication. Great job.

Reviewer #4 (Remarks to the Author):

The authors have done a great job clarifying and addressing my previous questions and comments. I have no further comments.